# A Unified Framework for Tabular Generative Modeling: Loss Functions, Benchmarks, and Improved Multi-objective Bayesian Optimization Approaches

**Minh H. Vu**                                                        *minh.vu@umu.se*
*Department of Diagnostics and Intervention*
*Umeå University*

**Daniel Edler**                                                      *daniel.edler@umu.se*
*Department of Physics*
*Umeå University*

**Carl Wibom**                                                        *carl.wibom@umu.se*
*Department of Diagnostics and Intervention*
*Umeå University*

**Tommy Löfstedt**                                                    *tommy.lofstedt@umu.se*
*Department of Computing Science*
*Umeå University*

**Beatrice Melin**                                                    *beatrice.melin@umu.se*
*Department of Diagnostics and Intervention*
*Umeå University*

**Martin Rosvall**                                                    *martin.rosvall@umu.se*
*Department of Physics*
*Umeå University*

**Reviewed on OpenReview:** *https://openreview.net/forum?id=RPZOEWOlzO*

## Abstract

Deep learning (DL) models require extensive data to achieve strong performance and generalization. Deep generative models (DGMs) offer a solution by synthesizing data. Yet current approaches for tabular data often fail to preserve feature correlations and distributions during training, struggle with multi-metric hyperparameter selection, and lack comprehensive evaluation protocols. We address this gap with a unified framework that integrates training, hyperparameter tuning, and evaluation. First, we introduce a novel correlation- and distribution-aware loss function that regularizes DGMs, enhancing their ability to generate synthetic tabular data that faithfully represents the underlying data distributions. Theoretical analysis establishes stability and consistency guarantees. To enable principled hyper-parameter search via Bayesian optimization (BO), we also propose a new multi-objective aggregation strategy based on iterative objective refinement Bayesian optimization (IORBO), along with a comprehensive statistical testing framework. We validate the proposed approach using a benchmarking framework with twenty real-world datasets and ten established tabular DGM baselines. The correlation-aware loss function significantly improves the synthetic data fidelity and downstream machine learning (ML) performance, while IORBO consistently outperforms standard Bayesian optimization (SBO) in hyper-parameter selection. The unified framework advances tabular generative modeling beyond isolated method improvements. Code is available at: https://github.com/vuhoangminh/TabGen-Framework.

# 1 Introduction

For a wide range of deep learning (DL) applications, large amounts of data are crucial to improve both model performance and generalization. The fast-paced advancements in deep generative modeling have opened exciting possibilities for data synthesis. Models trained on images and text effectively learn probability distributions over complex data and generate high-quality, realistic samples (Karras et al., 2021; Team et al., 2023). This success on structured data has fueled a surge in deep generative model (DGM)-based methods (Goodfellow et al., 2014) for tabular data generation in recent years.

Yet tabular data poses unique challenges that resist direct transfer of techniques from other domains. Unlike images or text, tabular data lacks clear structure and contains mixed continuous and discrete variables with complex interactions, imbalances, and non-linear relationships. Recent hybrid approaches have explored combining diffusion processes with flow-based models and gradient-boosted trees to boost synthesis fidelity on tabular benchmarks (Jolicoeur-Martineau et al., 2024; Zein & Urvoy, 2022). However, these methods still rely on unguided likelihood or adversarial objectives and do not explicitly enforce key statistics such as feature correlations or higher-order moments.

This mismatch between tabular data complexity and current generative approaches cascades across the entire modeling lifecycle. Existing deep neural network (DNN)-based generative models often struggle to reliably capture correlations and other statistical dependencies in tabular data—sometimes failing to approximate even basic statistics such as the mean and variance—particularly in limited-data settings (Xu et al., 2019). Current approaches to improve downstream machine learning (ML) analyses focus primarily on addressing data imbalance (Xu et al., 2019; Sun et al., 2023; Zhao et al., 2021) while neglecting the equally crucial role of feature distributions and correlations.

The cascade extends to hyper-parameter optimization. While Bayesian optimization (BO) is widely used, standard approaches like standard Bayesian optimization (SBO) are ill-suited to aggregating the multiple heterogeneous metrics required for synthetic data evaluation. Combining metrics with different ranges and units, such as classification accuracy and regression error, via simple averaging can overweight individual objectives and yield suboptimal parameter selections.

Finally, the cascade undermines evaluation itself, where rigorous assessment remains fragmented. Existing methods often suffer from limited evaluation scopes that focus on narrow metric subsets, making it difficult to assess model performance across the complexities of diverse datasets. This evaluation gap obscures whether apparent improvements reflect genuine advances or artifacts of selective testing.

The central problem is the absence of a unified framework that addresses tabular generative modeling across its full lifecycle: training, hyper-parameter tuning, and evaluation. Current approaches tackle these stages independently, missing opportunities for integrated solutions that could amplify improvements at each step.

To address this gap, we propose a comprehensive unified framework that tackles training, hyper-parameter tuning, and evaluation as interconnected challenges. First, we introduce a novel correlation- and distribution-aware loss function for DGMs designed to enforce statistical properties that existing generative models fail to capture reliably. Second, we develop iterative objective refinement Bayesian optimization (IORBO), which aggregates multiple evaluation metrics through ranking to resolve inconsistencies caused by metrics with different units or scales. Third, we establish a comprehensive benchmarking framework that evaluates synthetic data across twenty datasets using statistical, regression, and classification metrics. By integrating these components, we create a unified pipeline where training improvements and robust hyper-parameter tuning work in concert with rigorous evaluation. The tight coupling between training, tuning, and evaluation improves statistical fidelity, robust optimization, and benchmarking rigor across diverse datasets.

In summary, we provide:

1. **A Correlation- and Distribution-Aware Loss Function:** We propose a custom correlation- and distribution-aware loss function that emphasizes the importance of feature correlations and distributions in tabular data. This custom loss function is used as an auxiliary regularization term, complementing the primary training objective. It significantly enhances the performance of DGMs,

    including generative adversarial network (GAN), variational auto-encoder (VAE), and denoising diffusion probabilistic model (DDPM), as demonstrated through extensive benchmark evaluations.

2. **Iterative Objective Refinement Bayesian Optimization:** We propose IORBO to aggregate multiple objectives through ranking, resolving inconsistencies caused by metrics with different units or scales.

3. **Benchmarking Framework for Synthetic Data Generation Algorithms:** We establish a comprehensive open-source benchmarking framework that includes twenty tabular datasets and various evaluation metrics based on statistical tests. This framework implements ten state-of-the-art tabular DGMs and supports extensions with additional methods and datasets.

The rest of the manuscript is organized as follows: Section 2 reviews related work on DGMs for tabular data and highlights recent developments motivating our approach. Section 3 presents the proposed correlation- and distribution-aware loss function (Section 3.1), the IORBO (Section 3.2), and the benchmarking framework (Section 3.3–3.6). Section 4 provides theoretical guarantees for the proposed loss functions, including stability analysis, error bounds, and generalized approximation guarantees. In Section 5, we describe the datasets and implementation details and training. Section 6 reports and discusses the results, including performance improvements and ablation studies. Finally, Section 7 concludes the paper and outlines future directions.

## 2 Related Work

Most existing methods to generate synthetic tabular data developed in the past decade model measurements in a table as a joint parametric density and then sample from that parametric model. Different models have been employed based on data characteristics: multivariate Gaussian (Frühwirth-Schnatter et al., 2018), Bayesian networks (Aviñó et al., 2018; Zhang et al., 2017), and copulas (Patki et al., 2016) for non-linearly correlated continuous variables. However, these methods rely on strong modeling assumptions and are limited in their ability to generalize beyond those assumptions. As a result, they often fail to capture the complex relationships present in real-world tabular data.

To overcome these limitations, recent approaches have turned to more expressive and flexible DGMs, such as VAEs (Kingma & Welling, 2013), diffusion models (Sohl-Dickstein et al., 2015; Kotelnikov et al., 2023), and GANs with their numerous extensions (Arjovsky et al., 2017; Gulrajani et al., 2017; Zhu et al., 2017; Yu et al., 2017), have made them very appealing for data representation. This appeal extends to generating tabular data, especially in the healthcare domain. For example, Yahi et al. (2017) leveraged GANs to create synthetic continuous time-series medical records, and Camino et al. (2018) proposed to generate discrete tabular healthcare data using GANs. `CTGAN` (Xu et al., 2019), `DP-CGANS` (Sun et al., 2023), and `CTAB-GAN` (Zhao et al., 2021) were proposed to address the complexities of mixed-type tabular data and to address challenges when generating realistic synthetic data, particularly for imbalanced datasets. More recently, `TabDDPM` (Kotelnikov et al., 2023), a diffusion model designed specifically for tabular data, offers the flexibility to incorporate various backbone architectures to model the reverse process. While these DGMs improve flexibility, they still struggle to capture the diverse variable types, imbalances, and intricate dependencies inherent in real-world tabular data.

Prior works in correlation and statistical regularization–most prominently CORAL by Sun et al. (2017)–focus on aligning second-order statistics (covariance) between source and target domains for unsupervised domain adaptation in computer vision. While effective for visual tasks, these methods typically assume continuous features and perform full covariance alignment, which can be costly both computationally and statistically in high-dimensional settings. By contrast, our correlation- and distribution-aware loss is tailored to tabular data with mixed continuous and discrete variables: it leverages variance- (diagonal covariance) based statistics for scalability, explicitly allows matching arbitrary higher-order moments (not just the second moment as in CORAL), and thus avoids the expense of estimating full covariance matrices while retaining the ability to capture richer distributional structure. Furthermore, our loss is integrated across multiple DGMs and evaluated on extensive downstream ML tasks and statistical metrics, providing theoretical guarantees alongside empirical improvements over standard baseline losses.

Beyond DGMs, several recent works have explicitly tackled the challenge of generating or imputing heterogeneous tabular data with mixed data types and missingness. Nazabal et al. (2020) proposed a variational autoencoder framework tailored for incomplete heterogeneous data, demonstrating strong performance in both imputation and generative tasks. Building on this direction, Ma et al. (2020) introduced `VAEM`, a DGM capable of capturing complex dependencies in mixed-type data, providing a principled approach for handling heterogeneous feature spaces. More recently, Peis et al. (2022) extended these ideas with deep hierarchical models combined with Hamiltonian Monte Carlo, achieving state-of-the-art results in missing data imputation and acquisition for structured heterogeneous datasets. These methods highlight the importance of explicitly modeling heterogeneous variable types, a direction complementary to our proposed correlation- and distribution-aware losses.

In parallel, BO has become an essential tool for hyper-parameter tuning in generative modeling. Classic methods such as the tree-structured parzen estimator approach (TPE; Bergstra et al., 2011) have been widely used for black-box optimization but often struggle with multi-metric objectives, treating metrics independently or aggregating them naively, which can bias hyper-parameter selection. More recent approaches improve multi-objective optimization and robustness, yet they still face challenges with heterogeneous, sometimes conflicting evaluation metrics common in tabular generative modeling. Our proposed IORBO addresses these issues by introducing a rank-based aggregation scheme that preserves metric relationships and supports fair, robust hyper-parameter tuning, directly tackling the sensitivity and correlation-ignorance of prior methods while improving model training and evaluation.

## 3 Methods

DGMs learn to map a random noise vector, denoted by $\boldsymbol{z}$, to an output sample. This allows them to generate new data instances that resemble the training data. DGMs have found various applications, *e.g.*, to generate images (Goodfellow et al., 2014; Karras et al., 2020), multi-modal medical images (Zhu et al., 2017), or vectors of tabular data (Xu et al., 2019; Sun et al., 2023). In this work, the focus was to generate tabular data with continuous and discrete variables.

### 3.1 A Correlation- and Distribution-Aware Loss Function

Let the training dataset be $\mathbf{X} = \{\boldsymbol{x}_i = (\boldsymbol{x}_i^{(c)}, \boldsymbol{x}_i^{(d)}) : \forall i \in \{1, \ldots, N\}\}$, where $N$ is the number of training samples. The $\boldsymbol{x}_i \in \mathbb{R}^m$ denotes the $i$-th training sample from $\mathbf{X}$, and $\boldsymbol{x}_i^{(c)}$ and $\boldsymbol{x}_i^{(d)}$ are continuous and discrete features, respectively. Let $p_{\tilde{\mathbf{X}}}$ be the learned probability density over the synthetic data, $\tilde{\boldsymbol{x}}$, such that $\tilde{\boldsymbol{x}} \in \mathbb{R}^m$ is a sample from the DGM, $\mathcal{G}$. Here, $\mathcal{G}$ is a learned mapping from a prior distribution $p(\boldsymbol{z})$ to the data space $p(\boldsymbol{x} \mid \boldsymbol{z})$.

*Correlation-aware loss function.* The correlation-aware loss function is defined as

$$\mathcal{L}_{\text{correlation}} = \frac{2}{m(m-1)} \sum_{j=1}^{m} \sum_{k=j+1}^{m} (\boldsymbol{g}_{j,k} - \tilde{\boldsymbol{g}}_{j,k})^2, \tag{1}$$

where $\boldsymbol{g}$ is the sample correlation over the real data and $\tilde{\boldsymbol{g}}$ is the sample correlation over the generated data, such that

$$\boldsymbol{g}_{j,k} = \frac{1}{N} \sum_{i=1}^{N} \frac{\boldsymbol{x}_{i,j} - \boldsymbol{\mu}_j}{\boldsymbol{\sigma}_j + \epsilon} \cdot \frac{\boldsymbol{x}_{i,k} - \boldsymbol{\mu}_k}{\boldsymbol{\sigma}_k + \epsilon}, \tag{2}$$

$$\tilde{\boldsymbol{g}}_{j,k} = \frac{1}{B} \sum_{i=1}^{B} \frac{\tilde{\boldsymbol{x}}_{i,j} - \tilde{\boldsymbol{\mu}}_j}{\tilde{\boldsymbol{\sigma}}_j + \epsilon} \cdot \frac{\tilde{\boldsymbol{x}}_{i,k} - \tilde{\boldsymbol{\mu}}_k}{\tilde{\boldsymbol{\sigma}}_k + \epsilon}, \tag{3}$$

with $B$ the size of the mini-batch used when training the DGM, and elements $\boldsymbol{x}_{i,j}$ and $\tilde{\boldsymbol{x}}_{i,j}$ belonging to vectors $\boldsymbol{x}_i \in \mathbf{X}$ and $\tilde{\boldsymbol{x}}_i \in \tilde{\mathbf{X}}$, respectively. A small positive value, $\epsilon = 1 \cdot 10^{-5}$, was added to the denominators of the correlation terms to avoid division by zero. The mean and standard deviation of the $j$-th column in a

tabular data set, $\mathbf{X}$, were estimated as

$$\boldsymbol{\mu}_j = \frac{1}{N} \sum_{i=1}^{N} \boldsymbol{x}_{i,j} \quad \text{and} \quad \boldsymbol{\sigma}_j = \sqrt{\frac{1}{N} \sum_{i=1}^{N} (\boldsymbol{x}_{i,j} - \boldsymbol{\mu}_j)^2}. \tag{4}$$

Similarly, $\tilde{\boldsymbol{\mu}}_j$ and $\tilde{\boldsymbol{\sigma}}_j$ were estimated as the mean and standard deviation of the generated data, $\{\tilde{\boldsymbol{x}}_i : \forall i \in \{1, \dots, B\}\}$. We discuss the theoretical guarantees for the correlation-aware loss function, including the stability analysis, in Section 4.1.

*Distribution-aware loss function.* The distribution-aware loss function integrates the strengths of the method of moments and maximum likelihood estimation (MLE) to align with the true distribution by capturing both statistical moments and likelihood properties in order to enhance the model's ability to learn accurate data representations (Pearson, 1936; Rice, 2007). Additionally, the choice of moments over distance-based metrics, such as Wasserstein, is motivated by their computational efficiency and stability, as lower-order moments provide a robust approximation of the distribution while avoiding the high computational cost associated with distance-based methods. To characterize the training data distribution, we employed the raw first and central second moments,

$$\mathcal{S}_j^{(1)} = \frac{1}{N} \sum_{i=1}^{N} \boldsymbol{x}_{i,j} = \boldsymbol{\mu}_j, \tag{5}$$

$$\mathcal{S}_j^{(2)} = \frac{1}{N} \sum_{i=1}^{N} (\boldsymbol{x}_{i,j} - \boldsymbol{\mu}_j)^2 = \boldsymbol{\sigma}_j^2, \tag{6}$$

and for $h \geq 3$ the standardized higher moments,

$$\mathcal{S}_j^{(h)} = \frac{1}{N} \sum_{i=1}^{N} \left( \frac{\boldsymbol{x}_{i,j} - \boldsymbol{\mu}_j}{\boldsymbol{\sigma}_j} \right)^h = \boldsymbol{\gamma}^h. \tag{7}$$

Similarly, the empirical moments were computed for the synthetic data, denoted as $\widetilde{\mathcal{S}}_j^{(1)}$, $\widetilde{\mathcal{S}}_j^{(2)}$, and $\widetilde{\mathcal{S}}_j^{(h)}$, again for $h \geq 3$. In this case, $B$ was used in place of $N$. Finally, the distribution loss was defined as

$$\mathcal{L}_{\text{distribution}} = \frac{1}{m} \sum_{j=1}^{m} \sum_{h=1}^{H} \frac{1}{h} \left( 1 - \frac{\widetilde{\mathcal{S}}_j^{(h)} + \epsilon}{\mathcal{S}_j^{(h)} + \epsilon} \right)^2, \tag{8}$$

where the number of moments, $H$, was treated as a hyper-parameter. Instead of making the moments equal, their quotient was made to be equal to one as a way to handle scale differences. By using a unified distribution-aware loss, we handle continuous and discrete variables in the same manner, simplifying the implementation and preventing imbalances that could arise from separate regularization terms for different data types. We discuss the theoretical guarantees for the distribution-aware loss function, including the numerical stability and consistency, in Section 4.2.

*Custom loss function for DGMs.* The correlation- and distribution-aware loss function was integrated into three DGMs: GAN, VAE, and DDPM. For GANs, the proposed loss function was incorporated into the generator's loss

$$\widetilde{\mathcal{L}}_G = \underbrace{\mathbb{E}_{\boldsymbol{z} \sim p_{\boldsymbol{z}}(\boldsymbol{z})} \left[ \log(1 - D(G(\boldsymbol{z}))) \right]}_{\mathcal{L}_G} + \alpha \mathcal{L}_{\text{correlation}} + \beta \mathcal{L}_{\text{distribution}}, \tag{9}$$

where $\mathcal{L}_G$ is the original GAN's generator loss, and $G$ and $D$ the generator and discriminator of the GAN, respectively. The hyper-parameters, $\alpha$ and $\beta$, controlled the influence of the correlation and distribution terms.

We extended the `TVAE` model (Xu et al., 2019) (a VAE designed for tabular data) with the proposed loss function

$$\widetilde{\mathcal{L}}_{\text{TVAE}} = \underbrace{\mathcal{L}_{\text{reconstruction}} + \mathcal{L}_{\text{KLD}}}_{\mathcal{L}_{\text{TVAE}}} + \alpha \mathcal{L}_{\text{correlation}} + \beta \mathcal{L}_{\text{distribution}}, \tag{10}$$

where $\mathcal{L}_{\text{TVAE}}$ is the original `TVAE`'s loss, and $\mathcal{L}_{\text{reconstruction}}$ and $\mathcal{L}_{\text{KLD}}$ are the reconstruction loss and the Kullback–Leibler (KL) regularization term, respectively.

For the diffusion model, `TabDDPM` (Kotelnikov et al., 2023), the proposed loss function was integrated into the total loss of the multinomial diffusions as

$$\widetilde{\mathcal{L}}_{\text{TabDDPM}} = \underbrace{\mathcal{L}_t^{\text{simple}} + \frac{\sum_{i \le C} L_t^i}{C}}_{\mathcal{L}_{\text{TabDDPM}}} + \alpha \mathcal{L}_{\text{correlation}}^{(d)} + \beta \mathcal{L}_{\text{distribution}}^{(d)} + \zeta \mathcal{L}_{\text{distribution}}^{(c)}, \tag{11}$$

where $\mathcal{L}_{\text{TabDDPM}}$ denotes the original `TabDDPM` loss, comprising the mean-squared error for the Gaussian diffusion term, $\mathcal{L}_t^{\text{simple}}$, and the KL divergence for all multinomial diffusion terms, $\sum_{i \le C} L_t^i / C$ (Kullback & Leibler, 1951).

Unlike other DGMs, `TabDDPM` handles continuous and discrete features separately. For continuous features, `TabDDPM` predicts the Gaussian noise added through a forward Markov process. For discrete features, it predicts their one-hot encoded representation. To align our proposed loss functions with this characteristic, we adapted the correlation and distribution loss functions, $\mathcal{L}_{\text{correlation}}^{(d)}$ and $\mathcal{L}_{\text{distribution}}^{(d)}$, to focus exclusively on discrete features. For continuous features, the Gaussian input noise is treated as the real data and the `TabDDPM`'s predicted noise component as the synthetic data. To encourage the model to capture the distribution of the predicted noise, we introduce a distributional loss term $\mathcal{L}_{\text{distribution}}^{(c)}$ for continuous features. A weighting parameter $\zeta$ is applied to this term to control its influence in the overall loss function, allowing to balance the importance of distributional alignment between real and predicted noise against other objectives.

It is important to note that the proposed correlation- and distribution-aware loss functions are incorporated as auxiliary regularization terms added to the primary training objective (*e.g.*, likelihood or adversarial loss), ensuring that the DGM remains guided by its standard optimization criterion while explicitly preserving correlations and distributional properties.

*Scope and applicability.* The proposed correlation- and distribution-aware loss functions are specifically designed for tabular data, where feature-level correlations and marginal distributions carry the most informative signals. While the losses themselves are general, applying them directly to raw pixels or text is less meaningful due to strong local structure and sequential dependencies. Nonetheless, the framework could be adapted to work on learned embeddings from other modalities, such as image or text encoders, which provide a structured representation suitable for correlation- and distribution-based regularization.

## 3.2 Iterative Objective Refinement Bayesian Optimization

Previous research on DNN often relied on tuning hyper-parameters based on a single metric or aggregating multiple metrics with varying units in SBO. For example, the objective function guiding the BO process could be the Dice score for medical segmentation (Vu et al., 2021), mean macro-accuracy for visual question answering (Vu et al., 2020), or metrics like F-score (classification) and $R$-squared (regression) evaluated with Catboost (Dorogush et al., 2018) on synthetic tabular data (Kotelnikov et al., 2023). A significant challenge in SBO arises from managing diverse metrics, such as those used in statistical evaluations and ML performance, that differ in units, complicating direct aggregation. This limitation can hinder the ability to fully capture trade-offs between different objectives. To overcome the issues associated with aggregating metrics with varying units in multi-objective SBO, we propose a ranking-based approach, named IORBO, to enhance BO performance.

To illustrate, consider optimizing a DGM. We define $\boldsymbol{y}_u$ as the vector comprising all evaluated metrics where $u \in \{1, \ldots, U\}$, with $U$ representing the number of samples used in the optimization. In the SBO, the

| **Algorithm 1** SBO | **Algorithm 2** IORBO |
|---|---|
| Initialize surrogate model | Initialize surrogate model |
| Initialize generative model $\mathcal{G}_i$ | Initialize generative model $\mathcal{G}_i$ |
| Suggest initial hyper-parameters $\Theta_1$ | Suggest initial hyper-parameters $\Theta_1$ |
| Build and train $\mathcal{G}_i$ | Build and train $\mathcal{G}_i$ |
| Perform evaluation to obtain $\boldsymbol{y}_1$ | Perform evaluation to obtain $\boldsymbol{y}_1$ |
| Compute $r_1 = f(\boldsymbol{y}_1)$ | Compute $r_1^{(1)} = g(\boldsymbol{y}_1 \mid \boldsymbol{y}_1)$ |
| Fit surrogate model with $(\Theta_1, r_1)$ | Fit surrogate model with $(\Theta_1, r_1^{(1)})$ |
| **for** $u \leftarrow 2$ to $U$ **do** | **for** $u \leftarrow 2$ to $U$ **do** |
|     Suggest $\Theta_u$ |     Suggest $\Theta_u$ |
|     Build and train $\mathcal{G}_i$ |     Build and train $\mathcal{G}_i$ |
|     Perform evaluation to obtain $\boldsymbol{y}_u$ and $r_u$ |     Perform evaluation to obtain $\boldsymbol{y}_u$ |
|     Update surrogate model with $(\Theta_u, r_u)$ |     Update ranks $\{r_1^{(u)}, r_2^{(u)}, \ldots, r_u^{(u)}\}$ based on $\{\boldsymbol{y}_1, \boldsymbol{y}_2, \ldots, \boldsymbol{y}_u\}$ |
| |     Fit surrogate model with revised samples $(\Theta_1, r_1^{(u)}), (\Theta_2, r_2^{(u)}), \ldots, (\Theta_u, r_u^{(u)})$ |
| **end for** | **end for** |
| **return** Optimal hyper-parameters $\Theta^*$ | **return** Optimal hyper-parameters $\Theta^*$ |

Comparison between SBO and IORBO algorithms.

objective function of sample $u$ is defined as $r_u = f(\boldsymbol{y}_u)$ where $f$ is an aggregation function. As outlined in Algorithm 1, the SBO holds $r_u$ constant throughout the optimization.

In contrast, IORBO defines the objective function as $r_u^{(p)}$, where $u \leq p$ and $p \in \{1, \ldots, U\}$ (see Algorithm 2). Here, $u$ represents the iteration where the objective is first generated, while $p$ denotes when it is updated, introducing iterative refinement into the process. In the IORBO, the objective function of sample $u$ is defined as $r_u^{(p)} = g(\boldsymbol{y}_u \mid \boldsymbol{y}_1, \boldsymbol{y}_2, \ldots, \boldsymbol{y}_p)$ where $g$ is a rank-based function. For example, at the second iteration, $\boldsymbol{y}_2$ is evaluated, then both $r_1^{(2)}$ and $r_2^{(2)}$ are computed. In the third iteration, $\boldsymbol{y}_3$ is added, allowing for the computation of $r_1^{(3)}$, $r_2^{(3)}$, and $r_3^{(3)}$, and so on. The objective functions are recalculated as the mean ranks of all generated samples, yielding $r_1^{(u)}, r_2^{(u)}, \ldots, r_u^{(u)}$ based on $\boldsymbol{y}_1, \boldsymbol{y}_2, \ldots, \boldsymbol{y}_u$. To compute the mean ranks, all data points that are generated by the IORBO for each evaluated metric are first ranked and then the average rank across metrics is calculated.

The objective function for the first set of hyper-parameters, $\Theta_1$, is iteratively updated: $r_1^{(1)} \to r_1^{(2)} \to \cdots \to r_1^{(U)}$. For the $\Theta_2$, we update: $r_2^{(2)} \to r_2^{(3)} \to \cdots \to r_2^{(U)}$, and so on. The surrogate model is simultaneously refitted with the revised samples, $(\Theta_1, r_1^{(u)}), (\Theta_2, r_2^{(u)}), \ldots, (\Theta_u, r_u^{(u)})$. IORBO incurs a slight additional cost for refitting the surrogate model with revised samples during the iterative refinement. However, this overhead is negligible compared to the overall computational cost. Apart from this refinement step, the process is essentially the same as SBO. For a numerical illustration, see Section 3.2.1.

### 3.2.1 Illustrative Example: SBO and IORBO in Practice Comparison

Table 1 and Table 2 illustrate the key difference between SBO and IORBO across three optimization iterations using a toy example with four evaluation metrics: $\boldsymbol{y} = \{a, b, c, d\}$. Each row in the tables corresponds to the metric values obtained by evaluating a set of hyper-parameters at a given iteration.

In Table 1, the objective function is computed as the mean of all metric values for each evaluated sample. As new samples are added, the objective values for previous samples remain fixed, meaning $r_1 = r_2 = r_3 = 1$ across all iterations. This is because the SBO applies a static aggregation to each sample independently, without revisiting or comparing across iterations.

In contrast, Table 2 shows how IORBO iteratively refines the objective function. At each iteration, metric values are ranked across all evaluated samples to compute the average rank per sample. This process results in dynamic objective values: $r_1^{(3)} \neq r_2^{(3)} \neq r_3^{(3)}$ and $r_1^{(1)} \neq r_1^{(2)} \neq r_1^{(3)}$. The key idea is that as more samples are evaluated, the ranking context changes, and thus the relative standing of earlier samples is updated to

Table 1: Example of SBO where the objective function is computed as the mean of all evaluated metrics.

| Metric / Sample | Iteration 1 | Iteration 2 | | Iteration 3 | | |
|---|---|---|---|---|---|---|
| | 1 | 1 | 2 | 1 | 2 | 3 |
| $a$ | 1 | 1 | 0.5 | 1 | 0.5 | 2.4 |
| $b$ | 1 | 1 | 2.5 | 1 | 2.5 | 0.2 |
| $c$ | 1 | 1 | 0.5 | 1 | 0.5 | 0.8 |
| $d$ | 1 | 1 | 0.5 | 1 | 0.5 | 0.6 |
| Objective function | $r_1 = 1$ | $r_1 = 1$ | $r_2 = 1$ | $r_1 = 1$ | $r_2 = 1$ | $r_3 = 1$ |

Table 2: Example of IORBO.

| Metric / Sample | Iteration 1 | Iteration 2 | | Iteration 3 | | |
|---|---|---|---|---|---|---|
| | 1 | 1 | 2 | 1 | 2 | 3 |
| $a$ | 1 | 1 | 0.5 | 1 | 0.5 | 2.4 |
| $b$ | 1 | 1 | 2.5 | 1 | 2.5 | 0.2 |
| $c$ | 1 | 1 | 0.5 | 1 | 0.5 | 0.8 |
| $d$ | 1 | 1 | 0.5 | 1 | 0.5 | 0.6 |
| Metric ranking / Sample | 1 | 1 | 2 | 1 | 2 | 3 |
| $a$ | 1 | 2 | 1 | 2 | 1 | 3 |
| $b$ | 1 | 1 | 2 | 2 | 3 | 1 |
| $c$ | 1 | 2 | 1 | 3 | 1 | 2 |
| $d$ | 1 | 2 | 1 | 3 | 1 | 2 |
| Objective function | $r_1^{(1)} = 1$ | $r_1^{(2)} = 1.75$ | $r_2^{(2)} = 1.25$ | $r_1^{(3)} = 2.5$ | $r_2^{(3)} = 1.5$ | $r_3^{(3)} = 2$ |

reflect the expanded information. For instance, the first sample has objective values $r_1^{(1)} = 1$, $r_1^{(2)} = 1.75$, and $r_1^{(3)} = 2.5$, demonstrating the refinement process.

This illustrative example highlights how IORBO uses cross-sample comparison to improve the fidelity of the optimization signal, allowing it to better differentiate between competing hyper-parameter configurations. In contrast, SBO may lead to flat or misleading optimization signals when metric values vary in scale or importance. This rank-based refinement in IORBO enables more informed and robust surrogate model updates throughout the optimization.

### 3.3 Evaluation

*Statistical similarity.* The statistical similarity evaluation focuses on how well the statistical properties of the real training data are preserved in the synthetic data. Inspired by a previous review study (Goncalves et al., 2020), we compared two aspects: (1) Individual variable distributions assess how closely the distributions of each variable in the real and synthetic data sets resemble each other; and (2) pairwise correlations reveal the differences in pairwise correlations between variables across the real and synthetic data (Step 1 in Figure 2).

We employed four key metrics to quantify how closely the real and synthetic data distributions resemble each other. (1) The KL divergence (Hershey & Olsen, 2007): This method quantifies the information loss incurred when approximating a true probability distribution with another one. (2) The Pearson's Chi-Square (CS) test (Pearson, 1992): This test focuses on categorical variables and assesses whether the distribution of categories in the synthetic data matches the distribution in the real data. (3) The Kolmogorov–Smirnov (KS) test (Massey Jr, 1951): This test is designed for continuous variables and measures the distance between the cumulative distribution functions (CDFs) of the real and synthetic data. (4) The dimension-wise probability (DWP): We leveraged the DWP (Armanious et al., 2020) to quantitatively assess the quality of the generated data. This metric evaluates how well the model captures the distribution of each individual class or variable.

To calculate the DWP metric, we compute the average distance between scatter points and a perfect diagonal line ($y = x$). Each scatter point represents either a class within a categorical variable or the mean value of a continuous variable.

To assess how effectively the synthetic data captures the inherent relationships between variables observed in the real data, we compare correlation coefficients between variable pairs. For continuous variables, we employ the widely-used Pearson correlation coefficient, calculated from both the real and synthetic data matrices. In the case of categorical variables, we leverage Cramer's V coefficient to quantify the association strength between each pair in both datasets (Frey, 2018).

While Pearson and Cramer's V handle continuous-continuous and categorical-categorical correlations, continuous-categorical dependencies are not explicitly measured. These can be partially captured by one-hot encoding categorical variables or indirectly via the distribution-aware loss. Addressing this explicitly remains a direction for future work.

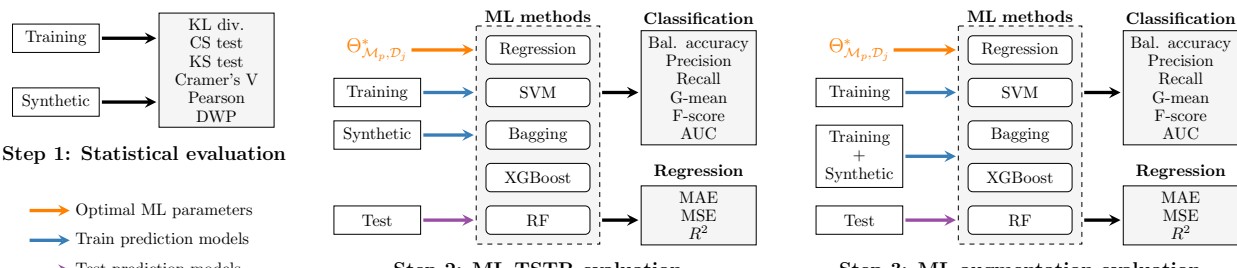

Figure 2: Evaluation pipeline. For dataset $\mathcal{D}_j$ and ML method $\mathcal{M}_p$, the optimal hyper-parameters, $\Theta^*_{\mathcal{M}_p, \mathcal{D}_j}$, were determined using five-fold cross-validation based on ML evaluation metrics (see Figure 4 in the Appendix).

*ML performance.* The ML performance evaluation is meant to enable researchers to leverage synthetic data when developing ML methods in two key areas: Train-Synthetic-Test-Real (TSTR) (Lu et al., 2023) and augmentation (see Figure 2 and Steps 2 and 3). In the TSTR task (Step 2 in Figure 2), the goal for ML methods trained on synthetic data was to achieve performance comparable or identical to those trained on real data.

In addition, we introduce the concept of an ML augmentation benchmark task (Step 3 in Figure 2), which, to our knowledge, represents the first systematic application of augmentation for evaluating tabular synthetic data. Here, models are trained on a combination of real and synthetic data, and the objective is to outperform models trained solely on real data. While data augmentation with synthetic samples is a well-established practice in machine learning, our contribution lies in formalizing it as a reproducible and domain-agnostic benchmark for assessing DGMs across diverse tabular datasets. By incorporating synthetic data in this standardized evaluation, models can learn from richer distributions, providing a practical measure of generative model utility.

To comprehensively evaluate the performance of trained ML models on imbalanced classification datasets, we employed a suite of metrics including balanced accuracy, precision, recall, geometric mean (G-mean), F-score, and area under the ROC curve (AUC). For regression, we used metrics focused on capturing regression error: mean absolute error (MAE), mean squared error (MSE), and the coefficient of determination, $R$-squared ($R^2$). This combined evaluation approach provides a nuanced understanding of model performance across both classification and regression tasks.

To assess the ML performance in both the TSTR and augmentation tasks, we split the experimental datasets into 80% training and 20% testing sets. First, we trained the DGMs on the real training data to produce synthetic data. The real testing set served a critical role in assessing the generalizability of trained ML models on unseen data. Subsequently, for TSTR, we trained various ML methods including logistic regression (LG), support vector machine (SVM), random forest (RF), bagging (bootstrap aggregating) on top of LG, and XGBoost independently on both the real and synthetic training sets. In the augmentation task, we

trained the same ML models independently on both the real training set and a combined set consisting of real training and synthetic data.

### 3.4 Hyper-parameter Search

Hyper-parameters play a pivotal role in tailoring ML methods and DGMs to specific datasets and achieving optimal performance. To systematically optimize the hyper-parameters, we employed BO, which was introduced in Section 3.2. Specifically, we considered both SBO (baseline) and the proposed IORBO variant for multi-objective aggregation. In practice, unless otherwise stated, all experiments are run with IORBO, while SBO is included only for comparison.

We implemented BO using the TPE algorithm (Bergstra et al., 2011) within the `Hyperopt`[1] library, which efficiently explores black-box functions to identify optimal hyper-parameter configurations. This approach enabled us to effectively navigate the complex hyper-parameter space and select suitable settings for each experiment.

We conducted two distinct tuning processes. First, each ML method used in ML TSTR and augmentation evaluation (Figure 2 and Step 2 and 3), was fine-tuned for each dataset using five-fold cross-validation on the ML evaluation metrics (Figure 4 in the Appendix). Second, we optimized the hyper-parameters for each combination of DGM, dataset, and loss function (Figure 3 in the Appendix).

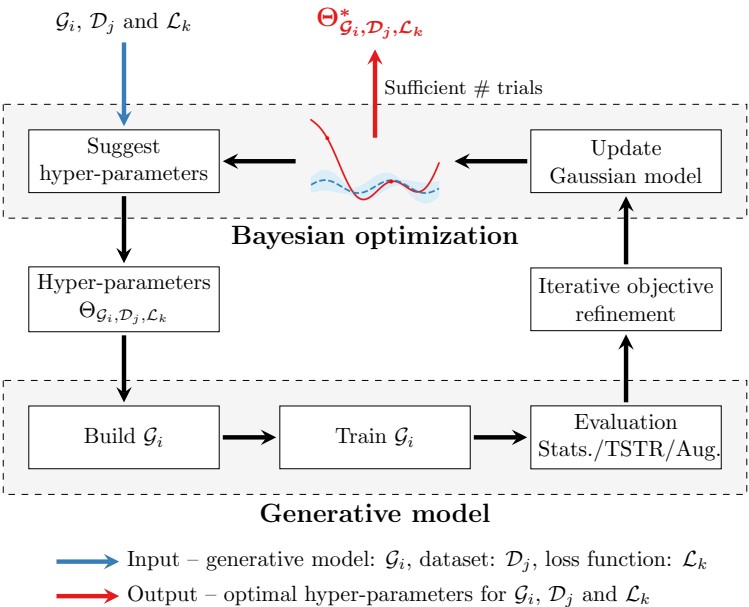

Figure 3: Hyper-parameter search for a single generative model.

### 3.5 Statistical Tests

In Table 3, the specifications are based on the commonly accepted interpretation of $p$-values in hypothesis testing. A $p$-value less than or equal to 0.01 ($p \leq 0.01$) indicates that the result is highly significant, meaning that the null hypothesis can be rejected with high confidence. A $p$-value between 0.01 and 0.05 ($0.01 < p \leq 0.05$) indicates significant results, where there is still a reasonable level of evidence against the null hypothesis, though not as strong as for the highly significant results. For $p$-values greater than 0.05, we consider the result not to be significant, indicating insufficient evidence to reject the null hypothesis.

---

[1]`https://hyperopt.github.io/hyperopt/`

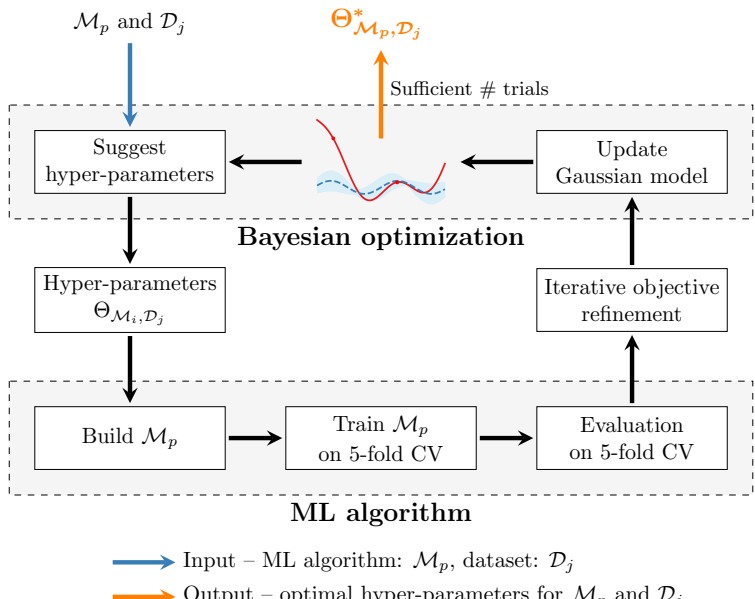

Figure 4: Hyper-parameter search for an ML algorithm.

Regarding the two-sided test, the Nemenyi post-hoc test used in our analysis is based on the Friedman test, which is a non-parametric test for repeated measures. The Nemenyi test performs pairwise comparisons between the groups following the Friedman test and is a two-sided test. This means that the test evaluates whether the differences between the groups are statistically significant in both directions, *i.e.*, it considers whether one group is significantly better or worse than another group.

To compare loss functions across DGMs and datasets, we used the Friedman test (Friedman, 1937; 1940) to rank the loss functions independently. For non-parametric analysis of repeated-measures data, the Friedman test offers an alternative to the widely used repeated-measures ANOVA (Fisher, 1919). We used the Friedman test with equivalence on two ML efficacy problems for test set predictions and statistical similarity between training and synthetic data (detailed in Section 3.6). Following Demšar (2006), we further explored significant differences between methods using the Nemenyi post-hoc test (Nemenyi, 1963). Table 3 shows the $p$-values divided into three positive and three negative differences.

Table 3: Ranges of $p$-values and specification obtained from statistical tests.

| Notation | Rank | Range of $p$-value | Specification |
|---|---|---|---|
| ++ | Better | $p \leq 0.01$ | Highly significantly better |
| + | Better | $0.01 < p \leq 0.05$ | Significantly better |
| 0 | Better | $p > 0.05$ | Not significantly better |
| 0 | Worse | $p > 0.05$ | Not significantly worse |
| − | Worse | $0.01 < p \leq 0.05$ | Significantly worse |
| −− | Worse | $p \leq 0.01$ | Highly significantly worse |

## 3.6 Benchmarking Framework

Figure 5 provides an overview of the proposed benchmarking framework consisting the following core components:

*Generative models.* DGMs are used to generate synthetic data. We evaluated six models. Three models that leverage conditional GANs for data synthesis: `CTGAN` (Xu et al., 2019), `CTAB-GAN` (Zhao et al., 2021), and `DP-CGANS` (Sun et al., 2023). A model that combines Gaussian Copula with the `CTGAN` architecture:

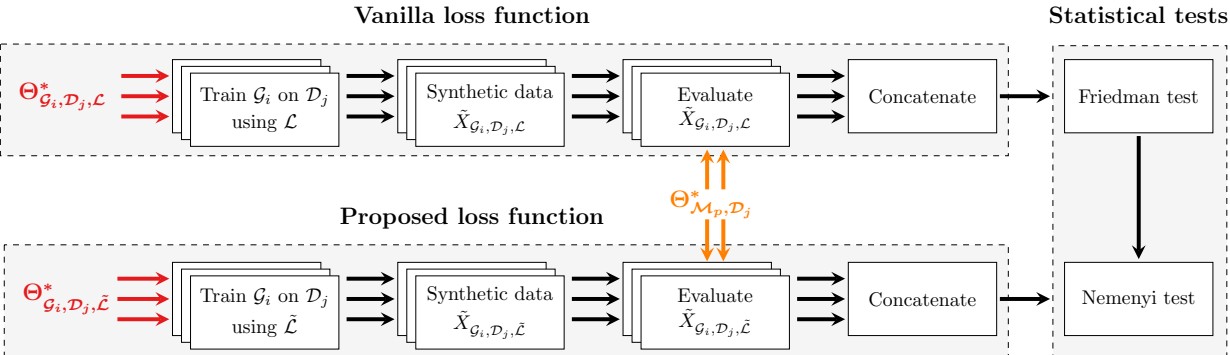

Figure 5: Proposed benchmarking framework. The $\mathcal{G}_i$, $\mathcal{D}_j$, $\mathcal{L}_k$ and $\mathcal{M}_p$ denote a DGM, a dataset, a loss function, and an ML method, respectively. $\Theta^*$ denotes the optimal set of hyper-parameters. See Figure 3 and Figure 4 in the Appendix to see how we determined $\Theta^*_{\mathcal{G}_i,\mathcal{D}_j,\mathcal{L}_k}$ and $\Theta^*_{\mathcal{M}_p,\mathcal{D}_j}$.

`CopulaGAN`. A model that utilizes VAEs (Kingma & Welling, 2013) for data generation: `TVAE` (Xu et al., 2019). Finally, a model that employs DDPM: `TabDDPM` (Kotelnikov et al., 2023). To explore the impact of the conditional element, we additionally evaluated versions of `CTGAN`, `CopulaGAN`, and `DP-CGANS` with conditioning disabled. We also used two backbones for `TabDDPM`: a simple multilayer perceptron (MLP) and a ResNet.

*Custom loss function.* During training, each evaluated DGM utilized either the custom loss function defined in Equation 9 (for GAN models), the one presented in Equation 10 (for `TVAE` model) or the one in Equation 11 (for `TabDDPM` model). We subsequently fixed $\alpha$, $\beta$, and $\zeta$ to specific values of 0 or a positive value, resulting in two different experiments: vanilla loss function ($\mathcal{L}$ with $\alpha = \beta = \zeta = 0$) and the proposed loss function, $\widetilde{\mathcal{L}}$, with at least one non-zero hyper-parameters.

*Statistical tests.* We used the Friedman test on all evaluated metrics, followed by the Nemenyi post-hoc test detailed in Section 3.5 for comparative analyses. These analyses can be divided into three categories: (1) General-purpose loss function assesses which loss function—between the vanilla (original loss function used in the evaluated DGM) and the proposed—performs better for general applications; (2) Dataset-specific loss determines which loss function is more effective for each evaluated dataset; and (3) Method-specific loss identifies the superior loss function for each evaluated DGM architecture. For each category, we based the evaluations on either statistical similarity, ML TSTR performance, ML augmentation performance, or a combination of evaluated metrics.

## 4 Theoretical guarantees

### 4.1 Correlation-aware loss function

We now analyze the theoretical properties of the proposed correlation-aware loss function, including its stability during optimization. This analysis provides justification for its robustness in practice. The following remark clarifies how we implement the correlation loss in a stochastic optimization setting.

*Remark* 4.1 (Optimization details). During the stochastic optimization of $\mathcal{L}_{\text{correlation}}$, the running estimates $\tilde{\boldsymbol{\mu}}_j$ (raw first moment) and $\tilde{\boldsymbol{\sigma}}_j$ (central second moment) are treated as fixed constants within each mini-batch. To ensure numerical stability and avoid division by zero, a small constant $\epsilon > 0$ is added to the denominator, *i.e.*, we use $\tilde{\boldsymbol{\sigma}}_j + \epsilon$.

To quantify how well matching the first $H$ moments controls the overall density approximation error, we now state our main result.

**Proposition 4.2** (Stability). *Assume the standardized synthetic data points*

$$Z_{i,k} := \frac{\tilde{\boldsymbol{x}}_{i,k} - \tilde{\boldsymbol{\mu}}_k}{\tilde{\boldsymbol{\sigma}}_k + \epsilon}$$

*have tails that decay at least sub-Gaussian; that is, there exist constants $K, \nu > 0$ such that*

$$\Pr\big(|Z_{i,k}| > t\big) \leq M\, e^{-\nu t^2},$$

*for all $t > 0$. Fix $\delta \in (0,1)$ and set*

$$t_\delta = \sqrt{\frac{1}{\nu} \ln\Big(\frac{M}{\delta}\Big)}.$$

*Then, with probability at least $1 - \delta$, the gradient of $\mathcal{L}_{\text{correlation}}$ with respect to any $\tilde{\boldsymbol{x}}_{i,j}$ satisfies*

$$\left| \frac{\partial \mathcal{L}_{\text{correlation}}}{\partial \tilde{\boldsymbol{x}}_{i,j}} \right| \leq \frac{8}{m(m-1)} \cdot \frac{t_\delta}{B\,\epsilon},$$

*where $B$ is the batch size and $\epsilon > 0$ is the smoothing constant.*

*Proof.* To analyze the gradient of $\mathcal{L}_{\text{correlation}}$ with respect to $\tilde{\boldsymbol{x}}_{i,j}$, consider a single pair $(j,k)$. The gradient can be expressed as:

$$\frac{\partial \mathcal{L}_{\text{correlation}}}{\partial \tilde{\boldsymbol{x}}_{i,j}} = \frac{4}{m(m-1)}(\boldsymbol{g}_{j,k} - \tilde{\boldsymbol{g}}_{j,k}) \cdot \frac{\partial \tilde{\boldsymbol{g}}_{j,k}}{\partial \tilde{\boldsymbol{x}}_{i,j}}.$$

Recall that the synthetic correlation term is given by:

$$\tilde{\boldsymbol{g}}_{j,k} = \frac{1}{B} \sum_{i=1}^{B} \frac{\tilde{\boldsymbol{x}}_{i,j} - \tilde{\boldsymbol{\mu}}_j}{\tilde{\boldsymbol{\sigma}}_j + \epsilon} \cdot \frac{\tilde{\boldsymbol{x}}_{i,k} - \tilde{\boldsymbol{\mu}}_k}{\tilde{\boldsymbol{\sigma}}_k + \epsilon}.$$

Since the statistics $\tilde{\boldsymbol{\mu}}_j$ and $\tilde{\boldsymbol{\sigma}}_j$ are treated as constants during stochastic optimization (Remark 4.1), the derivative simplifies to:

$$\frac{\partial \tilde{\boldsymbol{g}}_{j,k}}{\partial \tilde{\boldsymbol{x}}_{i,j}} = \frac{1}{B} \cdot \frac{1}{\tilde{\boldsymbol{\sigma}}_j + \epsilon} \cdot \frac{\tilde{\boldsymbol{x}}_{i,k} - \tilde{\boldsymbol{\mu}}_k}{\tilde{\boldsymbol{\sigma}}_k + \epsilon}.$$

Taking the absolute value and using the property $|ab| = |a||b|$, we obtain:

$$\left| \frac{\partial \mathcal{L}_{\text{correlation}}}{\partial \tilde{\boldsymbol{x}}_{i,j}} \right| = \frac{4}{m(m-1)} |\boldsymbol{g}_{j,k} - \tilde{\boldsymbol{g}}_{j,k}| \cdot \left| \frac{1}{B(\tilde{\boldsymbol{\sigma}}_j + \epsilon)} \cdot \frac{\tilde{\boldsymbol{x}}_{i,k} - \tilde{\boldsymbol{\mu}}_k}{\tilde{\boldsymbol{\sigma}}_k + \epsilon} \right|$$

$$\leq \frac{4}{m(m-1)} |\boldsymbol{g}_{j,k} - \tilde{\boldsymbol{g}}_{j,k}| \cdot \frac{1}{B(\tilde{\boldsymbol{\sigma}}_j + \epsilon)} \cdot \left| \frac{\tilde{\boldsymbol{x}}_{i,k} - \tilde{\boldsymbol{\mu}}_k}{\tilde{\boldsymbol{\sigma}}_k + \epsilon} \right|.$$

Here, the inequality follows because the absolute value of a product is the product of the absolute values. Since the correlations are bounded (i.e., $|\tilde{\boldsymbol{g}}_{j,k}| \leq 1$ and $|\boldsymbol{g}_{j,k}| \leq 1$, implying $|\boldsymbol{g}_{j,k} - \tilde{\boldsymbol{g}}_{j,k}| \leq 2$) and by the bounded data assumption we have

$$\left| \frac{\partial \mathcal{L}_{\text{correlation}}}{\partial \tilde{\boldsymbol{x}}_{i,j}} \right| \leq \frac{4}{m(m-1)} \cdot 2 \cdot \frac{1}{B(\tilde{\boldsymbol{\sigma}}_j + \epsilon)} \cdot |Z_{i,k}|.$$

By the sub-Gaussian tail bound,

$$\Pr\left(|Z_{i,k}| > t_\delta\right) \leq M e^{-\nu\, t_\delta^2} = \delta.$$

Thus with probability at least $1 - \delta$ we have $|Z_{i,k}| \leq t_\delta$, and noting that $\tilde{\boldsymbol{\sigma}}_j + \epsilon \geq \epsilon > 0$, it follows that

$$\left| \frac{\partial \mathcal{L}_{\text{correlation}}}{\partial \tilde{\boldsymbol{x}}_{i,j}} \right| \leq \frac{8}{m(m-1)} \cdot \frac{t_\delta}{B\,\epsilon}.$$

This completes the proof. $\qquad\square$

### 4.2 Distribution-aware loss function

We now establish theoretical guarantees for the proposed distribution-aware loss function.

**Assumption 4.3** (High-Probability Moment Matching)**.** For any confidence level $1 - \delta$, there exist constants $B_h(\delta) \geq 0$ such that, with probability at least $1 - \delta$,

$$\left|\hat{\mu}_h - \mu_h\right| \leq B_h(\delta) \quad \text{for all } h = 0, 1, \ldots, H.$$

**Proposition 4.4** (Numerical stability)**.** *Under Assumption 4.3, for any fixed confidence level, $1 - \delta$, there exist constants $B_h(\delta) > 0$ such that, with probability at least $1 - \delta$, the empirical moments satisfy*

$$\left|\mathcal{S}_j^{(h)}\right| \leq B_h(\delta) \quad and \quad \left|\widetilde{\mathcal{S}}_j^{(h)}\right| \leq B_h(\delta) \quad \forall j \in \{1, \ldots, m\}, \ h = 1, \ldots, H.$$

*Then for each feature index $j$ and moment order $h$, the partial derivative of*

$$\mathcal{L}_{\text{distribution}} = \frac{1}{m} \sum_{j=1}^{m} \sum_{h=1}^{H} \frac{1}{h} \left( 1 - \frac{\widetilde{\mathcal{S}}_j^{(h)} + \epsilon}{\mathcal{S}_j^{(h)} + \epsilon} \right)^2$$

*with respect to the synthetic moment $\widetilde{\mathcal{S}}_j^{(h)}$ obeys, with probability at least $1 - \delta$,*

$$\left| \frac{\partial \mathcal{L}_{\text{distribution}}}{\partial \widetilde{\mathcal{S}}_j^{(h)}} \right| \leq \frac{4 B_h(\delta)}{h \epsilon^2}.$$

Note: *We do not allow $\epsilon$ to go to zero. Instead, we treat $\epsilon$ as a small constant that maintains $\mathcal{S}_j^{(h)} + \epsilon \geq \epsilon$. That way, the bound remains finite and numerically reasonable.*

*Proof.* Recall that the distribution-aware loss function is defined as:

$$\mathcal{L}_{\text{distribution}} = \frac{1}{m} \sum_{j=1}^{m} \sum_{h=1}^{H} \frac{1}{h} \left( 1 - \frac{\widetilde{\mathcal{S}}_j^{(h)} + \epsilon}{\mathcal{S}_j^{(h)} + \epsilon} \right)^2,$$

where $\epsilon > 0$ is a smoothing term that ensures numerical stability by preventing division by zero.

To analyze the stability of $\mathcal{L}_{\text{distribution}}$, we compute its gradient with respect to the synthetic moment $\widetilde{\mathcal{S}}_j^{(h)}$:

$$\frac{\partial \mathcal{L}_{\text{distribution}}}{\partial \widetilde{\mathcal{S}}_j^{(h)}} = \frac{2}{h} \left( 1 - \frac{\widetilde{\mathcal{S}}_j^{(h)} + \epsilon}{\mathcal{S}_j^{(h)} + \epsilon} \right) \cdot \left( -\frac{1}{\mathcal{S}_j^{(h)} + \epsilon} \right).$$

The smoothing term $\epsilon > 0$ ensures that the denominator $\mathcal{S}_j^{(h)} + \epsilon$ is bounded away from zero. Taking the absolute value of this partial derivative, we obtain:

$$\left| \frac{\partial \mathcal{L}_{\text{distribution}}}{\partial \widetilde{\mathcal{S}}_j^{(h)}} \right| = \frac{2}{h} \cdot \frac{\left| 1 - \frac{\widetilde{\mathcal{S}}_j^{(h)} + \epsilon}{\mathcal{S}_j^{(h)} + \epsilon} \right|}{\mathcal{S}_j^{(h)} + \epsilon} = \frac{2}{h} \cdot \frac{\left| \mathcal{S}_j^{(h)} - \widetilde{\mathcal{S}}_j^{(h)} \right|}{\left( \mathcal{S}_j^{(h)} + \epsilon \right)^2}.$$

Under the high-probability bound in the Proposition 4.4, $|\mathcal{S}_j^{(h)}| \leq B_h(\delta)$ and $|\widetilde{\mathcal{S}}_j^{(h)}| \leq B_h(\delta)$, we have

$$\left| \mathcal{S}_j^{(h)} - \widetilde{\mathcal{S}}_j^{(h)} \right| \leq 2 B_h(\delta), \quad \mathcal{S}_j^{(h)} + \epsilon \geq \epsilon.$$

Therefore, with probability at least $1 - \delta$,

$$\left| \frac{\partial \mathcal{L}_{\text{distribution}}}{\partial \widetilde{\mathcal{S}}_j^{(h)}} \right| \leq \frac{2}{h} \cdot \frac{2 \, B_h(\delta)}{\epsilon^2} = \frac{4 \, B_h(\delta)}{h \, \epsilon^2}.$$

Since the bound depends only on $h$, $\epsilon$, and $B_h(\delta)$—all of which are independent of the feature index $j$—this implies that the partial derivatives are uniformly bounded across all $j$ and $h$. This completes the proof.

$\square$

*Remark* 4.5 (Empirical moment consistency). After training on $N$ real samples, we can draw $B$ synthetic samples from the learned model to estimate any feature moment. By the Law of Large Numbers:

$$\frac{1}{N} \sum_{i=1}^{N} (x_{i,j} - \mu_j)^h \xrightarrow[N \to \infty]{P} \mathbb{E}\left[ (X_j - \mu_j)^h \right], \quad \frac{1}{B} \sum_{i=1}^{B} (\tilde{x}_{i,j} - \tilde{\mu}_j)^h \xrightarrow[B \to \infty]{P} \mathbb{E}\left[ (X_j - \mu_j)^h \right].$$

Since the model's moments cannot typically be computed in closed form, we estimate them by averaging over $B$ synthetic samples. Larger values of $B$ lead to more accurate moment estimates. In practice, setting $B$ comparable to or larger than $N$ (*e.g.*, $B \in [N, 10N]$) typically yields sufficiently accurate moment estimates.

## 5 Experiments

### 5.1 Datasets

Two datasets come from the UCI Machine Learning Repository (Dua & Graff, 2017) (`Adult` and `News`) and feature tabular structures with separate columns for attributes and labels. Thirteen additional datasets were preprocessed and shared by Kotelnikov et al. (2023) including `Abalone`, `Buddy`, `California`, `Cardio`, `Churn2`, `Diabetes-ML`, `Gesture`, `Higgs-Small`, `House-16h`, `Insurance`, `King`, `Miniboone`, and `Wilt`. We sourced the remaining datasets from Kaggle[2] (`Credit`, `Diabetes`, `Balanced Diabetes`, and `House`). To investigate the proposed method's behavior on high-dimensional binary data as in (Xu et al., 2019), we transformed the Modified National Institute of Standards and Technology database (MNIST) dataset (LeCun & Cortes, 2010). Specifically, we binarized the original $28 \times 28$ images, converted each sample into a 784-dimensional vector, and added a label column. The images were then resized to $12 \times 12$, reducing them to 144-dimensional vectors. We refer to this dataset as `MNIST12`.

Table 4 provides a comprehensive overview of the datasets evaluated in this study. It includes a diverse set of datasets, encompassing various data types and tasks to thoroughly test the proposed methods. The datasets range from small, specialized datasets like `Diabetes-ML` with 768 rows and 8 continuous variables, to large, extensive datasets such as `Credit` with 277 640 rows and 29 continuous variables. Tasks represented include regression, binary classification, and multiclass classification, showcasing the breadth of application scenarios covered. For instance, `Abalone` and `California` are used for regression tasks, while `Adult`, `Cardio`, and `Churn2` are employed for binary classification tasks. Multiclass classification tasks are represented by datasets such as `Buddy` and `MNIST12`.

Additionally, the datasets exhibit a range of characteristics in terms of the number of continuous and discrete variables. For example, `Gesture` has a high number of continuous variables (32) with no discrete variables, whereas `Diabetes` features a substantial number of discrete variables (21) with no continuous variables. The varied nature of these datasets allows for a robust evaluation of the proposed methods across different types of data and tasks, providing insights into their generalizability and effectiveness. The inclusion of datasets with different characteristics, such as `Higgs-Small` with 28 continuous variables and `MNIST12` with 144 discrete variables, ensures a comprehensive assessment of performance and applicability.

Our benchmarking framework evaluates the proposed methods across twenty diverse real-world tabular datasets, spanning a range of scales to assess performance under varying conditions. These include small-scale datasets with thousands of rows, as well as mid- to large-scale ones, such as the Credit dataset with

---

[2]`https://www.kaggle.com/datasets`

Table 4: Description of experimented datasets.

| Dataset | #Rows | #Continuous | #Discrete | Task |
|---|---|---|---|---|
| Abalone | 4 177 | 7 | 1 | Regression |
| Adult | 48 813 | 6 | 8 | Binclass |
| Buddy | 18 834 | 4 | 5 | Multiclass |
| California | 20 640 | 8 | 0 | Regression |
| Cardio | 70 000 | 5 | 6 | Binclass |
| Churn2 | 10 000 | 7 | 4 | Binclass |
| Credit | 277 640 | 29 | 0 | Binclass |
| Diabetes | 234 245 | 0 | 21 | Binclass |
| Diabetes-ML | 768 | 8 | 0 | Binclass |
| Diabetes Bal. | 69 515 | 0 | 21 | Binclass |
| Gesture | 9 873 | 32 | 0 | Multiclass |
| Higgs-Small | 98 049 | 28 | 0 | Binclass |
| House | 21 613 | 10 | 8 | Regression |
| House-16h | 22 784 | 16 | 0 | Regression |
| Insurance | 1 338 | 3 | 3 | Regression |
| King | 21 613 | 17 | 3 | Regression |
| Miniboone | 130 064 | 50 | 0 | Binclass |
| MNIST12 | 70 000 | 0 | 144 | Multiclass |
| News | 39 644 | 45 | 14 | Regression |
| Wilt | 4 839 | 5 | 0 | Binclass |

approximately 277 000 rows. This selection allows us to test the methods on datasets representative of practical scenarios, though we note that even larger datasets (*e.g.*, millions of rows) are common in industrial applications and warrant future exploration.

## 5.2 Implementation Details and Training

*Implementation Details and Training.* We implemented all DGMs (`CTGAN`, `CTAB-GAN`, `DP-CGANS`, `CopulaGAN`, `TVAE`, and `TabDDPM`) and the proposed losses using PyTorch 1.13. To ensure replicability, we maintained the DGMs' original framework structures and adopted the model parameters specified in their publications. We disabled conditional elements within evaluated DGMs by reimplementing their data samplers. This modification removed the conditional vector from the training process, effectively transforming them into unconditional DGMs. For all DGMs, we employed the Adam optimizer (Kingma & Ba, 2015). We used the proposed IORBO approach introduced in Section 3.2 to fine-tune the hyper-parameters in two tuning processes (Section 3.4).

The experiments ran on a high-performance computing cluster equipped with NVIDIA A100 Tensor Core graphical processing units (GPUs) (40GB RAM each) and Intel(R) Xeon(R) Gold 6338 CPUs (256GB DDR4 RAM). Training time per model varied significantly by dataset and DGM, ranging from one hour to two weeks.

To accelerate the ML performance evaluation, we used the `cuML` library (Raschka et al., 2020). This library provides a Python API largely compatible with `scikit-learn` (Pedregosa et al., 2011) and allows seamless execution of traditional tabular ML tasks on GPUs. We used `scikit-learn` for classification and regression metrics, `scipy` for statistical evaluation metrics, and `scikit-posthocs` for the statistical tests, ensuring consistency throughout the evaluation process.

## 6 Results and Discussion

**Loss function.** To analyze the performance of the proposed loss function against the vanilla version, we employed the proposed benchmarking framework (Section 3.6) across four key tasks: statistical evaluation (Stat.), TSTR evaluation, augmentation evaluation (Aug.), and a comprehensive evaluation (Comp.) combining all three. The statistical tests evaluated the performance of the proposed loss function compared to the vanilla loss. In addition, we define the *win rate* as the proportion of evaluated metrics where the proposed loss function exceeds the vanilla loss function, relative to the total number of metrics assessed. A win rate of 1 indicates that the proposed loss function performed better than the vanilla version across all evaluated metrics, while a value of 0 signifies that it performed worse in every metric. A win rate greater than 0.5 indicates that the proposed loss function was "better" more often than it was "worse." We also report standard errors for each metric, estimated from 1 000 bootstrap rounds.

It is important to note that throughout the reported experiments, the "proposed loss function" corresponds to the vanilla training objective augmented with the auxiliary correlation- and distribution-aware regularizer. This ensures that the results reflect improvements attributable to the added regularization term rather than replacing the underlying optimization criterion (see Section 3.1). Furthermore, unless otherwise stated, all experiments are conducted using the proposed IORBO for hyper-parameter search, with SBO included solely as a baseline for comparison.

Table 5: Results of the Nemenyi post-hoc test and win rate (with standard error in parentheses) comparing the proposed against the vanilla loss function on all DGMs and datasets. Loss functions were evaluated for statistical similarity (Stat.), TSTR, augmentation (Aug.), and a comprehensive evaluation (Comp.) combining all metrics. For details on $p$-value ranges, refer to Table 3.

| | Statistical Tests | | | | Win Rate | | | |
|---|---|---|---|---|---|---|---|---|
| Comparison | Stat. | TSTR | Aug. | Comp. | Stat. | TSTR | Aug. | Comp. |
| Proposed vs. Vanilla | 0 | ++ | ++ | ++ | 0.484 (0.012) | **0.611 (0.007)** | **0.551 (0.007)** | **0.567 (0.004)** |

*General-purpose loss function.* Table 5 presents the results of a comprehensive analysis comparing the performance of the proposed loss function against the vanilla loss function across all DGMs and datasets. The table highlights the influence of loss function selection for general purposes. First, the two loss functions performed statistically similarly (zero (0) in the "Stat." column in Table 5). However, this metric does not fully capture performance in downstream tasks. In contrast, in the ML TSTR evaluation, the proposed loss function significantly outperformed the vanilla version, with a win rate of 0.611 and a standard error of 0.007, suggesting that the proposed loss function better captures the complexities of real-world tabular data during synthetic data generation. Similarly, the augmentation evaluation consistently favored the proposed loss function (win rate 0.551), demonstrating its ability to enhance the performance of predictive models trained on a mix of real and synthetic data. Finally, the comprehensive evaluation (win rate 0.567), which combined all prior evaluations, continues this trend, indicating the proposed loss function's potential to improve model generalizability. A possible reason for this superiority is that the proposed loss function provides a regularizing effect, which reduces overfitting on unseen data and positions it as a strong candidate for general-purpose use in generative modeling tasks.

*Method-specific loss function.* Table 6 compares the performance of the proposed loss function against the vanilla loss functions across all datasets and different DGM selections. Models denoted with an asterisk (*) have disabled conditioning. For most models, the proposed loss function demonstrates significant improvements in ML TSTR performance and augmentation effectiveness. For instance, CTGAN, CTGAN*, CopulaGAN, and DP-CGANS consistently show highly significant gains (++) in TSTR, augmentation, and comprehensive evaluation. For example, DP-CGANS* achieved the highest win rate across the TSTR metric, 0.798, indicating that the proposed loss function significantly enhanced its ability to generate synthetic data that boosts downstream ML performance. Interestingly, the statistical similarity (Stat.) evaluation reveals no significant differences between the proposed and vanilla loss functions for most models, suggesting that both loss functions perform similarly in terms of generating synthetic data that statistically match the

Table 6: Results of the Nemenyi post-hoc test and win rate (with standard error in parentheses) comparing the proposed against the vanilla loss function across various DGMs on all datasets. Evaluations include TSTR, augmentation (Aug.), statistical similarity (Stat.), and a comprehensive measure (Comp.) combining all evaluated metrics. Models denoted with an asterisk (*) have disabled conditioning. For details on $p$-value ranges, refer to Table 3.

| | Statistical Tests | | | | Win Rate | | | |
|---|---|---|---|---|---|---|---|---|
| Method | Stat. | TSTR | Aug. | Comp. | Stat. | TSTR | Aug. | Comp. |
| CTGAN | 0 | ++ | ++ | ++ | 0.478 (0.034) | **0.639 (0.020)** | **0.583 (0.021)** | **0.593 (0.014)** |
| CTGAN* | 0 | ++ | ++ | ++ | 0.459 (0.036) | **0.726 (0.018)** | **0.611 (0.020)** | **0.640 (0.013)** |
| TVAE | 0 | 0 | ++ | ++ | **0.519 (0.034)** | 0.501 (0.021) | **0.593 (0.020)** | **0.543 (0.013)** |
| CopulaGAN | 0 | ++ | + | ++ | 0.491 (0.033) | **0.633 (0.020)** | **0.547 (0.022)** | **0.577 (0.013)** |
| CopulaGAN* | 0 | ++ | + | ++ | 0.447 (0.034) | **0.684 (0.019)** | **0.554 (0.020)** | **0.595 (0.013)** |
| DP-CGANS | 0 | ++ | ++ | ++ | 0.500 (0.051) | **0.669 (0.028)** | **0.683 (0.028)** | **0.651 (0.018)** |
| DP-CGANS* | 0 | ++ | 0 | ++ | **0.587 (0.054)** | **0.798 (0.023)** | **0.538 (0.030)** | **0.656 (0.019)** |
| CTAB-GAN | −− | −− | 0 | −− | 0.391 (0.033) | 0.418 (0.020) | 0.497 (0.020) | 0.448 (0.014) |
| TABDDPM-MLP | 0 | ++ | 0 | ++ | **0.516 (0.035)** | **0.617 (0.020)** | 0.482 (0.021) | **0.545 (0.014)** |
| TABDDPM-ResNet | 0 | + | 0 | 0 | **0.512 (0.033)** | **0.547 (0.021)** | 0.487 (0.021) | **0.517 (0.013)** |

real data distributions. However, the `CTAB-GAN` model stands out as an exception, showing a statistically significant decrease ($−−$) in performance across most evaluations when using the proposed loss function. This result suggests that the `CTAB-GAN` may require a more specialized loss function or optimization strategy to fully benefit from the proposed approach. The comprehensive evaluation (Comp.), which combines all three metrics, underscores the effectiveness of the proposed loss function on eight out of ten evaluated DGMs. Models including `CTGAN`, `CopulaGAN`, `DP-CGANS`, and their non-conditioned variants, consistently outperform the vanilla loss function with win rates exceeding 0.5. These results imply that the proposed loss function offers a well-rounded improvement across various aspects of synthetic data generation, specifically in terms of enhancing ML utility and model augmentation performance.

*Dataset-specific loss function.* Table 7 compares the proposed loss function to the vanilla loss function across various datasets on all DGMs. The results demonstrate the effectiveness of the proposed loss function across diverse datasets. The statistical tests reveal that the proposed loss function achieves statistically significant improvements in TSTR performance for 14 out of 20 datasets, as indicated by the total count of ($+$) and ($++$). Additionally, the proposed loss function exhibits a consistent advantage in augmentation (Aug.). Specifically, datasets such as `Insurance` and `MNIST12` show marked improvements in win rates (0.7). Conversely, the proposed loss function shows variable performance in statistical similarity (Stat.) across datasets. While it significantly improves TSTR and augmentation tasks for many datasets, its impact on statistical similarity is less consistent, with some datasets like `Cardio`, `Higgs-Small`, and `Miniboone` exhibiting inferior results compared to the vanilla loss function. From Table 7 we see that the proposed loss function demonstrates significant improvement over the vanilla loss function in 15 out of 20 datasets, as indicated by the comprehensive evaluation (Comp.) in the statistical tests column. Among the remaining datasets, four show no significant difference (0) and only one shows a statistically significant disadvantage ($−$).

**Bayesian optimization method.** The performance of the IORBO was compared to that of the SBO using two aggregation methods: mean and median aggregation. For each dataset, we fine-tuned each generative model (DGM) across two loss functions and employed three distinct BO approaches. To assess the performance of these methods, statistical tests were conducted, focusing on comparing the effectiveness of each approach. The results are summarized in Table 8, which presents the win rates and standard errors, as well as the outcomes of the Nemenyi post-hoc test that compares the methods shown in the rows against those in the columns. The findings from the Nemenyi post-hoc test clearly show that the IORBO significantly outperforms both SBO-Mean and SBO-Median. Specifically, the win rates for the IORBO compared to SBO-Mean and SBO-Median were 0.591 and 0.561, respectively. These results demonstrate that the IORBO is not only more effective but also more robust in handling datasets with metrics that have different units, which often pose challenges in optimization tasks. The consistent superiority of IORBO highlights its potential as a reliable

Table 7: Results of the Nemenyi post-hoc test and win rate (with standard error in parentheses) comparing the proposed against the vanilla loss function across various datasets on all evaluated DGMs. Evaluations include TSTR, augmentation (Aug.), statistical similarity (Stat.), and a comprehensive measure (Comp.) combining all three. For details on $p$-value ranges, refer to Table 3.

| | Statistical Tests | | | | Win Rate | | | |
|---|---|---|---|---|---|---|---|---|
| Dataset | Stat. | TSTR | Aug. | Comp. | Stat. | TSTR | Aug. | Comp. |
| Abalone | 0 | ++ | −− | 0 | **0.594 (0.059)** | **0.633 (0.043)** | 0.375 (0.045) | **0.523 (0.027)** |
| Adult | 0 | ++ | 0 | ++ | **0.538 (0.052)** | **0.622 (0.025)** | **0.553 (0.025)** | **0.582 (0.017)** |
| Buddy | 0 | ++ | 0 | ++ | 0.500 (0.051) | **0.607 (0.022)** | **0.552 (0.022)** | **0.570 (0.014)** |
| California | 0 | 0 | 0 | + | 0.500 (0.045) | **0.583 (0.044)** | **0.583 (0.043)** | **0.566 (0.027)** |
| Cardio | − | ++ | ++ | ++ | 0.387 (0.054) | **0.577 (0.027)** | **0.590 (0.027)** | **0.560 (0.019)** |
| Churn2 | 0 | ++ | 0 | + | 0.500 (0.059) | **0.637 (0.025)** | 0.460 (0.026) | **0.543 (0.018)** |
| Credit | 0 | 0 | 0 | + | 0.500 (0.057) | **0.544 (0.027)** | **0.554 (0.030)** | **0.543 (0.018)** |
| Diabetes | 0 | ++ | 0 | ++ | 0.413 (0.031) | **0.620 (0.027)** | **0.533 (0.027)** | **0.557 (0.018)** |
| Diabetes-ML | 0 | ++ | 0 | ++ | 0.469 (0.057) | **0.719 (0.029)** | 0.479 (0.033) | **0.584 (0.020)** |
| Diabetes Bal. | 0 | ++ | 0 | ++ | 0.438 (0.032) | **0.717 (0.022)** | **0.538 (0.025)** | **0.605 (0.016)** |
| Gesture | 0 | 0 | 0 | 0 | **0.609 (0.056)** | **0.562 (0.032)** | 0.450 (0.029) | **0.518 (0.021)** |
| Higgs-Small | − | + | ++ | ++ | 0.359 (0.055) | **0.575 (0.031)** | **0.635 (0.032)** | **0.576 (0.021)** |
| House | 0 | ++ | ++ | ++ | 0.438 (0.049) | **0.667 (0.039)** | **0.667 (0.038)** | **0.618 (0.025)** |
| House-16h | 0 | 0 | 0 | 0 | 0.500 (0.044) | 0.442 (0.044) | 0.500 (0.046) | 0.477 (0.027) |
| Insurance | 0 | ++ | ++ | ++ | 0.494 (0.052) | **0.693 (0.037)** | **0.700 (0.037)** | **0.654 (0.025)** |
| King | 0 | ++ | 0 | ++ | **0.519 (0.055)** | **0.673 (0.039)** | **0.567 (0.039)** | **0.599 (0.026)** |
| Miniboone | − | −− | 0 | − | 0.359 (0.054) | 0.402 (0.031) | **0.512 (0.026)** | 0.446 (0.020) |
| MNIST12 | 0 | ++ | ++ | ++ | 0.484 (0.040) | **0.756 (0.023)** | **0.700 (0.024)** | **0.699 (0.016)** |
| News | + | ++ | 0 | ++ | **0.612 (0.052)** | **0.607 (0.040)** | **0.553 (0.042)** | **0.587 (0.024)** |
| Wilt | 0 | 0 | 0 | 0 | 0.469 (0.056) | **0.538 (0.025)** | **0.552 (0.026)** | **0.536 (0.016)** |

and broadly applicable BO method, suggesting it could be a valuable tool in a wide range of optimization tasks involving diverse types of data and models.

Table 8: Results of the Nemenyi post-hoc test and win rate (with standard error in parentheses) comparing the row to column method. For details on $p$-value ranges, refer to Table 3.

| | Statistical Tests | | | Win Rate | | |
|---|---|---|---|---|---|---|
| BO method | SBO-Mean | SBO-Median | IORBO | SBO-Mean | SBO-Median | IORBO |
| SBO-Mean | | −− | −− | | 0.461 (0.004) | 0.409 (0.004) |
| SBO-Median | ++ | | −− | **0.539 (0.004)** | | 0.439 (0.004) |
| IORBO | ++ | ++ | | **0.591 (0.004)** | **0.561 (0.004)** | |

**Bayesian optimization computational cost.** Figure 6 compares the per-iteration computational cost of IORBO and SBO across different numbers of evaluated metrics. Each iteration measures the time required to update the surrogate model (see Algorithms 1 and 2). The analysis shows that both IORBO and SBO have the same asymptotic computational complexity, scaling linearly with the number of metrics. As shown in the timing comparison, execution times for both methods largely overlap across metric counts of 1, 10, 30, and 100, with observed differences remaining negligible. Thus, IORBO achieves its improved optimization performance without incurring additional computational cost in practice.

**Ablation studies.** The ablation study results presented in Table 9 evaluate the impact of different loss function variants combined with two optimization strategies: SBO and IORBO (denoted IOR). In our experiments, the loss function variants are defined as follows: V represents the vanilla loss; C denotes the addition of the correlation loss; D indicates the addition of the distribution loss; and CD corresponds to the combination of both correlation and distribution losses (*i.e.*, the proposed loss function). The table is organized with comparisons **vs. Mean** and **vs. Median** to succinctly present the performance metrics.

In the **vs. Mean** comparison, adding the correlation loss (C) to the vanilla setting (SBO-V) increases the win rate from 0.500 to 0.563, making it the most impactful individual component. In contrast, adding only

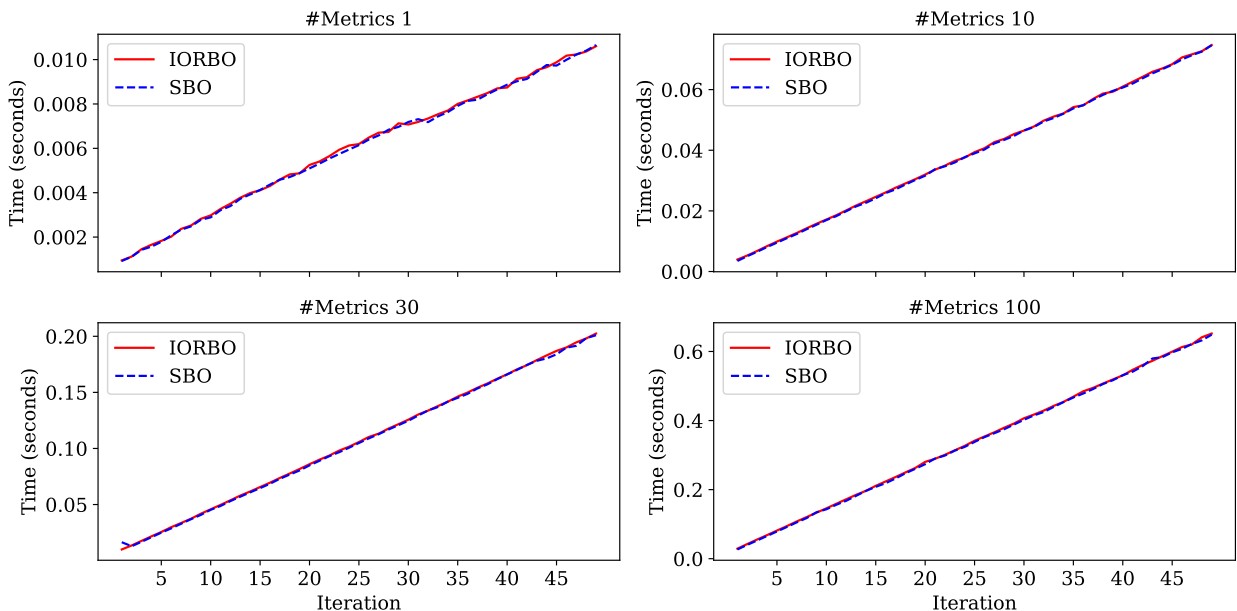

Figure 6: Per-iteration computational time of IORBO versus standard SBO across different numbers of evaluated metrics. Both methods show nearly identical timings, confirming that IORBO introduces no significant computational overhead.

the distribution loss (D) results in a smaller improvement from 0.500 to 0.508. However, the distribution loss should not be underestimated—it targets the overall data distribution, ensuring that the synthetic data not only capture pairwise correlations but also match higher-order statistical properties. When both loss components are combined in SBO-CD, the win rate reaches 0.542, indicating that the distribution loss provides complementary information that can enhance robustness and generalization, even though its impact is most pronounced when used in tandem with the correlation loss.

The effect of IOR-based optimization is even more substantial. Replacing SBO with IOR (IOR-V) boosts the win rate from 0.500 to 0.572, already surpassing most SBO variants except SBO-C. Further adding correlation loss (IOR-C) significantly improves performance, increasing the win rate from 0.572 to 0.636. Although adding only the distribution loss (IOR-D) results in a modest increase to 0.581, its contribution becomes crucial when combined with the correlation loss: the full combination (IOR-CD) achieves a win rate of 0.623, which, while slightly lower than IOR-C alone, offers more consistent performance across both **vs. Mean** and **vs. Median** evaluations. Notably, IOR-CD maintains a stable win rate of 0.522 in both comparisons, suggesting that the distribution loss enhances the overall robustness and consistency of the optimization process.

A similar pattern emerges in the **vs. Median** results. The addition of C to SBO-V improves performance from 0.500 to 0.558, while adding D results in a smaller gain to 0.514. The combined loss SBO-CD achieves 0.534, again slightly lower than SBO-C, which underscores that while the correlation loss is the dominant factor in boosting win rates, the distribution loss contributes to a more reliable performance. When switching to IOR, IOR-V increases the win rate from 0.500 to 0.546, and adding C further boosts performance to 0.599, making it the best-performing individual term. Although adding D alone results in a slight drop to 0.543, the full IOR-CD configuration achieves 0.586—again, slightly lower than IOR-C alone but with the added benefit of improved generalization across evaluation criteria.

Overall, these results highlight three key findings. First, among the two loss components, the correlation loss (C) has the strongest impact on performance. Second, while the distribution loss (D) provides only modest gains when used alone, it plays a crucial complementary role when combined with C, ensuring that both pairwise relationships and higher-order statistical properties are effectively captured. Third, IOR-based

Table 9: Ablation study results of the Nemenyi post-hoc test and win rate (with standard error in parentheses) comparing row to column methods. The table presents performance across different configurations using baseline methods with SBO and IORBO (denoted IOR). The loss function variations are as follows: V represents the vanilla loss, C adds the correlation loss, D adds the distribution loss, and CD adds both distribution and correlation losses, forming the proposed loss function. Both Mean and Median aggregation strategies are evaluated. For details on $p$-value ranges, refer to Table 3.

| | Statistical Tests | | | | | | | |
| --- | --- | --- | --- | --- | --- | --- | --- | --- |
| **vs. Mean** | SBO-V | IOR-V | SBO-C | IOR-C | SBO-D | IOR-D | SBO-CD | IOR-CD |
| SBO-V | | −− | −− | −− | 0 | −− | −− | −− |
| IOR-V | ++ | | ++ | −− | ++ | 0 | ++ | −− |
| SBO-C | ++ | −− | | −− | ++ | −− | 0 | −− |
| IOR-C | ++ | ++ | ++ | | ++ | ++ | ++ | 0 |
| SBO-D | 0 | −− | −− | −− | | −− | −− | −− |
| IOR-D | ++ | 0 | ++ | −− | ++ | | ++ | −− |
| SBO-CD | ++ | −− | 0 | −− | ++ | −− | | −− |
| IOR-CD | ++ | ++ | ++ | 0 | ++ | ++ | ++ | |

| | Win Rate | | | | | | | |
| --- | --- | --- | --- | --- | --- | --- | --- | --- |
| **vs. Mean** | SBO-V | IOR-V | SBO-C | IOR-C | SBO-D | IOR-D | SBO-CD | IOR-CD |
| SBO-V | | 0.428 (0.005) | 0.437 (0.005) | 0.364 (0.005) | 0.492 (0.005) | 0.419 (0.005) | 0.458 (0.005) | 0.377 (0.005) |
| IOR-V | 0.572 (0.005) | | **0.538 (0.005)** | 0.451 (0.005) | **0.579 (0.005)** | **0.526 (0.005)** | **0.547 (0.005)** | 0.427 (0.005) |
| SBO-C | **0.563 (0.004)** | 0.462 (0.005) | | 0.400 (0.005) | **0.555 (0.005)** | 0.457 (0.005) | **0.520 (0.005)** | 0.399 (0.005) |
| IOR-C | **0.636 (0.005)** | **0.549 (0.005)** | **0.600 (0.005)** | | **0.639 (0.005)** | **0.564 (0.005)** | **0.614 (0.005)** | 0.478 (0.005) |
| SBO-D | **0.508 (0.005)** | 0.421 (0.005) | 0.445 (0.005) | 0.361 (0.005) | | 0.425 (0.005) | 0.455 (0.005) | 0.375 (0.005) |
| IOR-D | **0.581 (0.005)** | 0.474 (0.005) | **0.543 (0.005)** | 0.436 (0.005) | **0.575 (0.005)** | | **0.544 (0.005)** | 0.429 (0.005) |
| SBO-CD | **0.542 (0.005)** | 0.453 (0.005) | 0.480 (0.005) | 0.386 (0.005) | **0.545 (0.005)** | 0.456 (0.005) | | 0.390 (0.005) |
| IOR-CD | **0.623 (0.005)** | **0.573 (0.005)** | **0.601 (0.005)** | **0.522 (0.005)** | **0.625 (0.005)** | **0.571 (0.005)** | **0.610 (0.005)** | |

| | Statistical Tests | | | | | | | |
| --- | --- | --- | --- | --- | --- | --- | --- | --- |
| **vs. Median** | SBO-V | IOR-V | SBO-C | IOR-C | SBO-D | IOR-D | SBO-CD | IOR-CD |
| SBO-V | | −− | −− | −− | 0 | −− | −− | −− |
| IOR-V | ++ | | 0 | −− | ++ | 0 | ++ | −− |
| SBO-C | ++ | 0 | | −− | ++ | ++ | ++ | −− |
| IOR-C | ++ | ++ | ++ | | ++ | ++ | ++ | 0 |
| SBO-D | 0 | −− | −− | −− | | −− | −− | −− |
| IOR-D | ++ | 0 | −− | −− | ++ | | 0 | −− |
| SBO-CD | ++ | − | −− | −− | ++ | 0 | | −− |
| IOR-CD | ++ | ++ | ++ | 0 | ++ | ++ | ++ | |

| | Win Rate | | | | | | | |
| --- | --- | --- | --- | --- | --- | --- | --- | --- |
| **vs. Median** | SBO-V | IOR-V | SBO-C | IOR-C | SBO-D | IOR-D | SBO-CD | IOR-CD |
| SBO-V | | 0.454 (0.005) | 0.442 (0.005) | 0.401 (0.005) | 0.486 (0.005) | 0.457 (0.005) | 0.466 (0.005) | 0.414 (0.005) |
| IOR-V | **0.546 (0.005)** | | 0.495 (0.005) | 0.451 (0.005) | **0.541 (0.005)** | **0.526 (0.005)** | **0.517 (0.005)** | 0.427 (0.005) |
| SBO-C | **0.558 (0.005)** | **0.505 (0.005)** | | 0.442 (0.005) | **0.558 (0.005)** | **0.514 (0.005)** | **0.524 (0.005)** | 0.450 (0.005) |
| IOR-C | **0.599 (0.005)** | **0.549 (0.005)** | **0.558 (0.005)** | | **0.593 (0.005)** | **0.564 (0.005)** | **0.562 (0.005)** | 0.478 (0.005) |
| SBO-D | **0.514 (0.005)** | 0.459 (0.005) | 0.442 (0.005) | 0.407 (0.005) | | 0.458 (0.005) | 0.474 (0.005) | 0.409 (0.005) |
| IOR-D | **0.543 (0.005)** | 0.474 (0.005) | 0.486 (0.005) | 0.436 (0.005) | **0.542 (0.005)** | | **0.506 (0.005)** | 0.429 (0.005) |
| SBO-CD | **0.534 (0.005)** | 0.483 (0.005) | 0.476 (0.005) | 0.438 (0.005) | **0.526 (0.005)** | 0.494 (0.005) | | 0.426 (0.005) |
| IOR-CD | **0.586 (0.005)** | **0.573 (0.005)** | **0.550 (0.005)** | **0.522 (0.005)** | **0.591 (0.005)** | **0.571 (0.005)** | **0.574 (0.005)** | |

optimization consistently enhances performance across all variants, with the best overall results achieved using the full combination (IOR-CD), which balances high win rates with robust generalization.

While our work focuses on tabular datasets, we note that the proposed losses could potentially be applied to embeddings derived from images or text. Such embeddings capture feature-level structure in a way that preserves meaningful correlations and distributions, making them suitable targets for correlation- and distribution-aware regularization. Exploring this direction represents a promising avenue for future research.

A key limitation of our current evaluation is that it focuses on datasets up to approximately 277,000 rows, whereas real-world tabular applications often involve millions of samples. This limitation is primarily due to computational time constraints rather than the scalability of the methods themselves. Specifically, training

DGMs and performing extensive hyper-parameter optimization across multiple baselines becomes prohibitively time-consuming on very large datasets. In our super-computing environment, we impose a 7-day maximum training time per experiment, which effectively caps the dataset sizes we can explore. Theoretically, our correlation- and distribution-aware loss functions scale linearly with the number of features ($m$) and batch size ($B$), with time complexities of $\mathcal{O}(m^2 B)$ for the correlation term (due to pairwise computations) and $\mathcal{O}(mHB)$ for the distribution term (where $H$ is the number of moments). Since these losses operate on mini-batches rather than the full dataset, they remain efficient as dataset size grows. Similarly, IORBO adds negligible overhead, with its ranking and surrogate refitting steps scaling, which is minor compared to DGM training costs. We expect our approach to behave robustly on larger scales: the losses should continue to enforce statistical fidelity effectively and may offer even greater benefits in high-volume scenarios where capturing complex dependencies is more challenging. Nonetheless, practical bottlenecks—particularly computational time, memory requirements for large batches, and the need for distributed training—may limit experiments on extremely large datasets.

## 7 Conclusion

We presented a unified framework for tabular generative modeling that integrates training, tuning, and evaluation. While tightly integrated, each component is also designed to be applied independently, making the framework modular. First, we introduced a novel correlation- and distribution-aware loss function designed as a regularizer for DGMs in tabular data synthesis, which outperforms the vanilla loss function across most DGMs. To ensure its robustness, we provided theoretical guarantees, including stability and consistency. The results suggest that the proposed loss function effectively captures the complexities of arbitrary DGMs. Future research could focus on developing a tailored loss function for the `CTAB-GAN` family to match the strong performance seen with other DGMs. Second, we introduced a novel IORBO approach that leverages rank-based aggregation to ensure more meaningful comparisons between multiple objectives with varying units, providing a more robust optimization process. Last, we developed a comprehensive benchmarking system evaluating statistical similarity, ML TSTR performance, and ML augmentation performance, with robust statistical tests, offering a valuable tool for future research.

## Acknowledgements

The computations and data handling were enabled by resources provided by the National Academic Infrastructure for Supercomputing in Sweden (NAISS) and the Swedish National Infrastructure for Computing (SNIC) at the Uppsala Multidisciplinary Center for Advanced Computational Science (UPPMAX), partially funded by the Swedish Research Council under grant agreements no. 2022-06725 and no. 2018-05973. This work was further supported by the WASP–DDLS postdoctoral grant Visualization and de-Identification of Biobank Data to Propel Precision Medicine Research (KAW 2023-03705), by the Swedish Cancer Foundation (24 3406 Pj 01 H; 21 1384 Pj 01 H), and by Umeå University Infrastructure funding (FS 2.1.6-1689-24).

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

# A    Hyper-Parameter Search Spaces

Table 10: Logistic Regression search space for classification dataset.

| Parameter | Distribution |
|-----------|--------------|
| C | LogUniform $(-4, 4)$ |
| max_iter | IntUniform $(50, 200)$ |
| l1_ratio | Uniform $(0, 1)$ |
| algorithm | {"svd", "eig", "qr", "svd-qr", "svd-jacobi"} |
| solver | {"newton-cg", "lbfgs", "liblinear", "sag", "saga"} |
| class_weight | {"balanced", None} |
| number of tuning iterations | 30 |

Table 11: ElasticNet search space for regression dataset.

| Parameter | Distribution |
|-----------|--------------|
| alpha | Uniform $(1, 10)$ |
| max_iter | IntUniform $(100, 2000)$ |
| l1_ratio | Uniform $(0, 1)$ |
| tol | LogUniform $(10^{-5}, 10^{-1})$ |
| fit_intercept | {True, False} |
| normalize | {True, False} |
| number of tuning iterations | 30 |

Table 12: Bagging for Logistic Regression search space for classification dataset.

| Parameter | Distribution |
|-----------|--------------|
| C | LogUniform $(-4, 4)$ |
| max_iter | IntUniform $(50, 200)$ |
| l1_ratio | Uniform $(0, 1)$ |
| algorithm | {"svd", "eig", "qr", "svd-qr", "svd-jacobi"} |
| solver | {"qn"} |
| class_weight | {"balanced", None} |
| number of tuning iterations | 30 |

For hyper-parameter search related to `TabDDPM`, please refer to the work by Kotelnikov et al. (2023).

Table 13: Bagging for ElasticNet search space for regression dataset.

| Parameter | Distribution |
|---|---|
| alpha | Uniform $(1, 10)$ |
| max_iter | IntUniform $(100, 2000)$ |
| l1_ratio | Uniform $(0, 1)$ |
| tol | LogUniform $(10^{-5}, 10^{-1})$ |
| fit_intercept | {True, False} |
| normalize | {True, False} |
| number of tuning iterations | 30 |

Table 14: SVM search space for classification dataset (LinearSVC).

| Parameter | Distribution |
|---|---|
| C | LogUniform $(0.1, 10)$ |
| max_iter | IntUniform $(100, 1500)$ |
| tol | LogUniform $(-5, -1)$ |
| penalty | {"hinge", "squared_hinge"} |
| loss | {True, False} |
| fit_intercept | {True, False} |
| penalized_intercept | {True, False} |
| class_weight | {"balanced", None} |
| number of tuning iterations | 30 |

Table 15: SVM search space for regression dataset (LinearSVR).

| Parameter | Distribution |
|---|---|
| C | LogUniform $(0.1, 10)$ |
| max_iter | IntUniform $(100, 1500)$ |
| tol | LogUniform $(-5, -1)$ |
| epsilon | Uniform $(0, 1)$ |
| fit_intercept | {True, False} |
| penalized_intercept | {True, False} |
| number of tuning iterations | 30 |

Table 16: RF search space for classification and regression dataset (RandomForestClassifier and RandomForestRegressor).

| Parameter | Distribution |
|---|---|
| n_estimators | IntUniform $(50, 500)$ |
| max_depth | IntUniform $(10, 100)$ |
| min_samples_split | IntUniform $(2, 20)$ |
| min_samples_leaf | IntUniform $(1, 20)$ |
| max_features | {"sqrt", "log2"} |
| number of tuning iterations | 30 |

Table 17: XGBoost search space for classification and regression dataset (XGBClassifier and XGBRegressor).

| Parameter | Distribution |
|---|---|
| n_estimators | IntUniform $(50, 500)$ |
| max_depth | IntUniform $(3, 15)$ |
| learning_rate | Uniform $(0.01, 0.3)$ |
| subsample | Uniform $(0.5, 1)$ |
| colsample_bytree | Uniform $(0.5, 1)$ |
| gamma | Uniform $(0, 5)$ |
| reg_alpha | Uniform $(0, 1)$ |
| reg_lambda | Uniform $(0, 1)$ |
| scale_pos_weight | Uniform $(1, 10)$ |
| number of tuning iterations | 30 |

Table 18: `CTGAN`, `CopulaGAN` and `DP-CGANS` search space.

| Parameter | Distribution |
|---|---|
| epochs | IntUniform $(100, 2\,000, 100)$ |
| batch_size | IntUniform $(500, 30\,000, 100)$ |
| embedding_dim | $\{32, 64, 128, 256\}$ |
| generator_dim | $\{32, 64, 128, 256\}$ |
| discriminator_dim | $\{32, 64, 128, 256\}$ |
| generator_learning_rate | Uniform $(10^{-5}, 10^{-3})$ |
| generator_decay | Uniform $(10^{-7}, 10^{-5})$ |
| discriminator_learning_rate | Uniform $(10^{-5}, 10^{-3})$ |
| discriminator_decay | Uniform $(10^{-7}, 10^{-5})$ |
| $\alpha$ | Uniform $(10^{-2}, 10^{4})$ |
| $\beta$ | Uniform $(10^{-10}, 10^{1})$ |
| number of moments | $\{1,2,3,4\}$ |
| number of tuning iterations | 30 |

Table 19: `TVAE` search space.

| Parameter | Distribution |
|---|---|
| epochs | IntUniform $(100, 2\,000, 100)$ |
| batch_size | IntUniform $(500, 30\,000, 100)$ |
| embedding_dim | $\{32, 64, 128, 256\}$ |
| compress_dims | $\{32, 64, 128, 256\}$ |
| decompress_dim | $\{32, 64, 128, 256\}$ |
| loss_factor | $\{0.25, 0.5, 1, 2, 4\}$ |
| l2scale | Uniform $(10^{-6}, 10^{-4})$ |
| $\alpha$ | Uniform $(10^{-2}, 10^{4})$ |
| $\beta$ | Uniform $(10^{-10}, 10^{1})$ |
| number of moments | $\{1,2,3,4\}$ |
| number of tuning iterations | 30 |

Table 20: `CTAB-GAN` search space.

| Parameter | Distribution |
|---|---|
| epochs | IntUniform $(100, 2\,000, 100)$ |
| batch_size | IntUniform $(500, 4\,000, 100)$ |
| test_ratio | $\{0.1, 0.2, 0.3, 0.4, 0.5\}$ |
| n_class_layer | $\{1, 2, 3, 4\}$ |
| class_dim | $\{32, 64, 128, 256\}$ |
| random_dim | $\{16, 32, 64, 128\}$ |
| num_channels | $\{16, 32, 64\}$ |
| $\alpha$ | Uniform $(10^{-2}, 10^4)$ |
| $\beta$ | Uniform $(10^{-10}, 10^1)$ |
| number of moments | $\{1,2,3,4\}$ |
| number of tuning iterations | 30 |

