# OpenReview forum: "A Unified Framework for Tabular Generative Modeling: Loss Functions, Benchmarks, and Improved Multi-objective Bayesian Optimization Approaches"
_TMLR — Accepted by TMLR_

### Review · Reviewer_QYER · 2025-07-23

**Summary Of Contributions:**

This paper presents three contributions aimed at improving tabular data generation with deep generative models (DGMs): (1) a correlation- and distribution-aware loss function to enhance the statistical fidelity of synthetic data, (2) a comprehensive benchmarking framework for evaluating synthetic tabular data using statistical and downstream ML metrics, and (3) a rank-based Bayesian optimization method (IORBO) for multi-metric hyperparameter tuning. While each component is potentially valuable, the connections between them are loosely integrated.

**Audience:**

Yes

**Broader Impact Concerns:**

While including an impact statement could be improved by briefly addressing risks of bias amplification in synthetic data, I don’t see major ethical concerns requiring changes.

**Claims And Evidence:**

No

**Requested Changes:**

- **Clarify better the connections between contributions**: Improve the flow of the paper so that the three contributions (loss, benchmarking, IORBO) are better integrated and motivated rather than appearing as isolated components.
- **Correct theoretical statements**: Rephrase or clarify inaccurate claims such as "neural networks cannot capture correlations."
- **Reframe the loss function**: Clearly describe the proposed loss as an auxiliary term, not the main optimization objective.
- **Revise the related work section**: Expand coverage to include prior work on:
  - Handling heterogeneous tabular data.
  - Bayesian optimization methods in generative modeling.
- **Clarify the novelty of the ML augmentation task**: Either justify its novelty or acknowledge prior work that uses real + synthetic data to improve ML performance.
- **Fix citation for KL Divergence**: Use the original reference [4] above.
- **Address clarity issues in methodology**: Avoid redundancy in Sections 3.2, 3.4, and 6. Clarify when IORBO is introduced and used.
- **Evaluate and report IORBO cost**: The paper claims the cost is negligible, but this should be justified with empirical or computational analysis.

**Strengths And Weaknesses:**

## Strengths
- The paper addresses an important problem in synthetic tabular data generation, where deep generative models often struggle.
- The proposed correlation- and distribution-aware loss function can be integrated into multiple DGM families (GANs, VAEs, DDPMs).
- The proposed rank-based Bayesian optimization method (IORBO) is an interesting contribution that addresses challenges in aggregating multi-metric objectives.

## Weaknesses
- The three main contributions (loss function, benchmarking framework, and IORBO) feel only loosely connected. The overall flow of the paper suffers from this disjointedness.
- Section 3.2 introduces IORBO as a new method, but Section 3.4 returns to standard BO as if introduced for the first time. In the experiments, IORBO is used exclusively, which adds to the confusion.
- Section 6 describing the experiment setup is repetitive and unclear — several paragraphs (e.g., the first and third) appear to describe the same content.
- There are theoretical inaccuracies, such as the suggestion that "neural networks cannot capture correlations" (Introduction), which is misleading.
- The proposed “loss function” is auxiliary and not an optimization objective in itself, though it is sometimes described as such.
- The claim of novelty for the “ML augmentation task” is questionable; using synthetic data alongside real data to improve model performance is a well-established technique.
- The related work section is vague and lacks coverage of key areas, especially existing methods for handling heterogeneous tabular data (e. g. [1-3]) and BO literature.
- In general, unless I misunderstood, the obtained win rates are not highly significant and should be interpreted with care. Only Table 7 shows significant results, which seem due to augmentations (well-known results).
- Unless I misinterpreted, Table 9 suggests that SBO-V (using standard losses with stochastic Bayesian optimization) often performs better than the proposed variant, which undermines the claimed benefits.

## References
[1] Nazabal, A., Olmos, P. M., Ghahramani, Z., & Valera, I. (2020). Handling incomplete heterogeneous data using vaes. Pattern Recognition, 107, 107501.

[2] Ma, C., Tschiatschek, S., Turner, R., Hernández-Lobato, J. M., & Zhang, C. (2020). Vaem: a deep generative model for heterogeneous mixed type data. Advances in Neural Information Processing Systems, 33, 11237-11247.

[3] Peis, I., Ma, C., & Hernández-Lobato, J. M. (2022). Missing data imputation and acquisition with deep hierarchical models and hamiltonian monte carlo. Advances in Neural Information Processing Systems, 35, 35839-35851.

[4] Kullback, Solomon, and Richard A. Leibler. "On information and sufficiency." The annals of mathematical statistics 22.1 (1951): 79-86.

---

> ### Author Response · Authors · 2025-09-30
> **Reply to Reviewer QYER [1]**
>
> **W1: The three main contributions (loss function, benchmarking framework, and IORBO) feel only loosely connected. The overall flow of the paper suffers from this disjointedness.**
>
> Thank you for raising this important point. Reviewer kBgG also highlighted a similar concern, and we are glad to re-clarify it here. We agree that, at first glance, the three contributions may appear distinct. However, the unifying theme of our work is enhancing the robustness and utility of tabular generative models across their full lifecycle (training → tuning → evaluation). Importantly, each contribution is motivated by concrete shortcomings in existing methods:
>
> -   Loss regularization: Current tabular deep generative models (DGMs) often fail to preserve correlations and higher-order statistics, which reduces the fidelity of generated data. Our correlation- and distribution-aware loss addresses this gap by explicitly capturing richer distributional properties, including both continuous and discrete variables.
> -   Hyperparameter tuning: Standard sequential Bayesian optimization (SBO) is not well-suited for the tabular generation setting, where models must be tuned against multiple heterogeneous metrics. This often leads to biased or unstable outcomes. Our IORBO method addresses this by providing a robust, fair approach to multi-metric hyperparameter optimization.
> -   Existing evaluation practices are fragmented and narrow, often focusing on a small subset of metrics. This makes it difficult to compare models fairly or assess robustness. Our unified evaluation framework broadens this scope and ensures rigorous, consistent assessment across diverse datasets.
>
> Taken together, these components form a coherent pipeline for advancing tabular generative modeling, rather than a collection of isolated techniques. While each method could in principle be applied in other domains, their integration is especially valuable here because tabular data generation requires improvements in training, tuning, and evaluation simultaneously.
>
> To make this unifying theme more explicit, we have revised the title, abstract, introduction, and conclusion. In specific, the revised title now reads:
> > A Unified Framework for Tabular Generative Modeling: Loss Functions, Benchmarks, and Improved Multi-objective Bayesian Optimization Approaches
>
> We have also expanded the introduction (highlighted in magenta in the revised draft):
> > [...]
> >
> > Yet tabular data poses unique challenges that resist direct transfer of techniques from other domains. Unlike images or text, tabular data lacks clear structure and contains mixed continuous and discrete variables with complex interactions, imbalances, and non-linear relationships. Recent hybrid approaches have explored combining diffusion processes with flow-based models and gradient-boosted trees to boost synthesis fidelity on tabular benchmarks (Jolicoeur-Martineau et al., 2024; Zein & Urvoy, 2022). However, these methods still rely on unguided likelihood or adversarial objectives and do not explicitly enforce key statistics such as feature correlations or higher-order moments.
> >
> > This mismatch between tabular data complexity and current generative approaches cascades across the entire modeling lifecycle. Existing deep neural network (DNN)-based generative models often struggle to reliably capture correlations and other statistical dependencies in tabular data—sometimes failing to approximate even basic statistics such as the mean and variance—particularly in limited-data settings (Xu et al., 2019). Current approaches to improve downstream machine learning (ML) analyses focus primarily on addressing data imbalance (Xu et al., 2019; Sun et al., 2023; Zhao et al., 2021) while neglecting the equally crucial role of feature distributions and correlations.
> >
> > The cascade extends to hyper-parameter optimization. While Bayesian optimization (BO) is widely used, standard approaches like standard Bayesian optimization (SBO) are ill-suited to aggregating the multiple heterogeneous metrics required for synthetic data evaluation. Combining metrics with different ranges and units, such as classification accuracy and regression error, via simple averaging can overweight individual objectives and yield suboptimal parameter selections.
> Finally, the cascade undermines evaluation itself, where rigorous assessment remains fragmented. Existing methods often suffer from limited evaluation scopes that focus on narrow metric subsets, making it difficult to assess model performance across the complexities of diverse datasets. This evaluation gap obscures whether apparent improvements reflect genuine advances or artifacts of selective testing.

---

> ### Author Response · Authors · 2025-09-30
> **Reply to Reviewer QYER [2]**
>
> > The central problem is the absence of a unified framework that addresses tabular generative modeling across its full lifecycle: training, hyper-parameter tuning, and evaluation. Current approaches tackle these stages independently, missing opportunities for integrated solutions that could amplify improvements at each step.
> To address this gap, we propose a comprehensive unified framework that tackles training, hyper-parameter tuning, and evaluation as interconnected challenges. First, we introduce a novel correlation- and distribution-aware loss function for DGMs designed to enforce statistical properties that existing generative models fail to capture reliably. Second, we develop iterative objective refinement Bayesian optimization (IORBO), which aggregates multiple evaluation metrics through ranking to resolve inconsistencies caused by metrics with different units or scales. Third, we establish a comprehensive benchmarking framework that evaluates synthetic data across twenty datasets using statistical, regression, and classification metrics. By integrating these components, we create a unified pipeline where training improvements and robust hyper-parameter tuning work in concert with rigorous evaluation. The tight coupling between training, tuning, and evaluation improves statistical fidelity, robust optimization, and benchmarking rigor across diverse datasets.
> >
> > [...]
>
> And in the Conclusion:
>
> > We presented a unified framework for tabular generative modeling that integrates training, tuning, and evaluation. While tightly integrated, each component is also designed to be applied independently, making the framework modular. [...]
>
>
>
> **W2: Section 3.2 introduces IORBO as a new method, but Section 3.4 returns to standard BO as if introduced for the first time. In the experiments, IORBO is used exclusively, which adds to the confusion.**
>
> Thank you for pointing this out. To remove the ambiguity, we have revised Section 3.4 "Hyper-parameter Search" as follows:
>
> > Hyper-parameters play a pivotal role in tailoring ML methods and DGMs to specific datasets and achieving optimal performance. To systematically optimize the hyper-parameters, we employed BO, which was
> introduced in Section 3.2. Specifically, we considered both SBO (baseline) and the proposed IORBO variant
> for multi-objective aggregation. In practice, unless otherwise stated, all experiments are run with IORBO, while SBO is included only for comparison.
> >
> > We implemented BO using the tree-structured parzen estimator approach (TPE) algorithm (Bergstra et al., 2011) within the Hyperopt library, which efficiently explores black-box functions to identify optimal hyper-parameter configurations. This approach enabled us to effectively navigate the complex hyper-parameter space and select suitable settings for each experiment. [...]
>
> Finally, in the Results and Discussion, we have added the following clarification to frame the reported results:
>
> > [...] Furthermore, unless otherwise stated, all experiments are conducted using the proposed IORBO for hyper-parameter search, with SBO included solely as a baseline for comparison.
>
>
>
> **W3: Section 6 describing the experiment setup is repetitive and unclear — several paragraphs (e.g., the first and third) appear to describe the same content.**
>
> Thank you. This was a mistake on our part — during re-organization of the manuscript we unintentionally left overlapping text in the first and second paragraphs. We have now removed these redundancies to improve clarity and ensure Section 6 is non-repetitive.

---

> ### Author Response · Authors · 2025-09-30
> **Reply to Reviewer QYER [3]**
>
> **W4: There are theoretical inaccuracies, such as the suggestion that "neural networks cannot capture correlations" (Introduction), which is misleading.**
>
> Thank you for raising this point. To clarify, our intention was not to suggest that neural networks are fundamentally incapable of capturing correlations. Rather, our claim is that existing DNN-based generative models for tabular data often fail to reliably preserve correlations and other statistical dependencies in practice, particularly when trained on limited data. This practical limitation has been noted in prior work; for example, Xu et al. (2019) report that DNN-based models may struggle to reproduce even basic statistics such as mean and variance, which further illustrates the challenges these models face in faithfully representing tabular distributions. We acknowledge that our original phrasing could be misread as overly strong, and have revised the Introduction accordingly. The revised text now states (highlighted in blue in the revised draft):
>
> > Existing deep neural network (DNN)-based generative models often struggle to reliably capture correlations and other statistical dependencies in tabular data—sometimes failing to approximate even basic statistics such as the mean and variance—particularly in limited-data settings (Xu et al., 2019). [...]
>
> This wording emphasizes practical limitations in current methods without suggesting a theoretical incapacity of neural networks.
>
>
> **W5: The proposed “loss function” is auxiliary and not an optimization objective in itself, though it is sometimes described as such.**
>
> You are correct that the proposed loss should be regarded as an auxiliary regularizer rather than a standalone optimization objective. We have revised the manuscript to clarify this distinction. In the Introduction, under the contributions, we now highlight that the loss complements the primary training objective by explicitly encouraging generative models to preserve correlations and distributional properties:
>
> [...] This custom loss function is used as an auxiliary regularization term, complementing the primary training objective [...]
>
> In the Methods section, we explicitly describe the correlation- and distribution-aware loss as an auxiliary regularization term added to the standard training objective, rather than as the sole optimization criterion (highlighted in blue text):
>
> > It is important to note that the proposed correlation- and distribution-aware loss functions are incorporated as auxiliary regularization terms added to the primary training objective (e.g., likelihood or adversarial loss), ensuring that the generative model remains guided by its standard optimization criterion while explicitly preserving correlations and distributional properties.
>
> Finally, in the Results and Discussion, we have added the following clarification to frame the reported results (highlighted in blue text):
>
> > It is important to note that throughout the reported experiments, the “proposed loss function” corresponds to the vanilla training objective  augmented with the auxiliary correlation- and distribution-aware regularizer. This ensures that the results reflect improvements attributable to the added regularization term rather than replacing the underlying optimization criterion (see Section 3.1). [...]

---

> ### Author Response · Authors · 2025-09-30
> **Reply to Reviewer QYER [4]**
>
> **W6: The claim of novelty for the “ML augmentation task” is questionable; using synthetic data alongside real data to improve model performance is a well-established technique.**
>
> Thank you for your comment. We agree that using real + synthetic data augmentation is a well-established concept in machine learning. Our contribution is not in proposing augmentation itself, but in systematically adapting augmentation as a standardized benchmark task for evaluating tabular generative models (DGMs). To the best of our knowledge, prior works on tabular DGMs have not employed augmentation explicitly as an evaluation paradigm. Instead, augmentation has been applied in isolated tasks or domains without being framed as a general benchmark to assess generative modeling quality.
>
> By integrating augmentation into our evaluation framework, we establish a reproducible and domain-agnostic benchmark that complements statistical and privacy metrics, thereby providing a more practical measure of utility. This benchmark allows for a fair comparison of DGMs across diverse datasets and model families, something that has not been systematically studied in earlier works.
>
> To clarify this point, we have updated Section 3.3 in the manuscript (highlighted in blue text):
>
> > In addition, we introduce the concept of an ML augmentation benchmark task (Step 3 in Figure 2), which, to our knowledge, represents the first systematic application of augmentation for evaluating tabular synthetic data. Here, models are trained on a combination of real and synthetic data, and the objective is to outperform models trained solely on real data. While data augmentation with synthetic samples is a well-established practice in machine learning, our contribution lies in formalizing it as a reproducible and domain-agnostic benchmark for assessing DGMs across diverse tabular datasets. By incorporating synthetic data in this standardized evaluation, models can learn from richer distributions, providing a practical measure of generative
> model utility.
>
>
> **W7: The related work section is vague and lacks coverage of key areas, especially existing methods for handling heterogeneous tabular data (e. g. [1-3]) and BO literature.**
>
> Thank you for the suggestion. We have extended the Related Work section as follows (highlighted in blue text):
>
> > [...] Beyond these models, several recent works have explicitly tackled the challenge of generating or imputing heterogeneous tabular data with mixed data types and missingness. Nazabal et al. (2020) proposed a variational autoencoder framework tailored for incomplete heterogeneous data, demonstrating strong performance in both imputation and generative tasks. Building on this direction, Ma et al. (2020) introduced VAEM, a DGM capable of capturing complex dependencies in mixed-type data, providing a principled approach for handling heterogeneous feature spaces. More recently, Peis et al. (2022) extended these ideas with deep hierarchical models combined with Hamiltonian Monte Carlo, achieving state-of-the-art results in missing data imputation and acquisition for structured heterogeneous datasets. These methods highlight the importance of explicitly modeling heterogeneous variable types, a direction complementary to our proposed correlation- and distribution-aware losses.
> >
> > In parallel, BO has become an essential tool for hyperparameter tuning in generative modeling. Classic methods such as the tree-structured parzen estimator approach (TPE; Bergstra et al., 2011) have been widely used for black-box optimization but often struggle with multi-metric objectives, treating metrics independently or aggregating them naively, which can bias hyperparameter selection. More recent approaches improve multi-objective optimization and robustness, yet they still face challenges with heterogeneous, sometimes conflicting evaluation metrics common in tabular generative modeling. Our proposed IORBO addresses these issues by introducing a rank-based aggregation scheme that preserves metric relationships and supports fair, robust hyperparameter tuning, directly tackling the sensitivity and correlation-ignorance of prior methods while improving model training and evaluation.

---

> ### Author Response · Authors · 2025-09-30
> **Reply to Reviewer QYER [5]**
>
> **W8: In general, unless I misunderstood, the obtained win rates are not highly significant and should be interpreted with care. Only Table 7 shows significant results, which seem due to augmentations (well-known results).**
>
> **W9: Unless I misinterpreted, Table 9 suggests that SBO-V (using standard losses with stochastic Bayesian optimization) often performs better than the proposed variant, which undermines the claimed benefits.**
>
> Thank you for comments W8 and W9. We believe there may have been some misunderstanding of how to interpret the tables and their captions. Let us clarify.
>
> - **Tables 5, 6, and 7** present results comparing **the proposed correlation- and distribution-aware loss function against the vanilla loss function**. While per-metric win rates can sometimes be modest, the Comprehensive (Comp.) measure—which aggregates all evaluated metrics—shows that the proposed loss function outperforms the vanilla loss function in most cases. Importantly, the Statistical Tests column (Comp.) demonstrates that the proposed loss function is *significantly better* than the vanilla loss function. This point is emphasized in Section *Results and Discussion*, where we explicitly note that the statistical test outcomes provide the strongest evidence in favor of our proposed loss.
>
> - **Tables 8 and 9** instead compare **row methods vs. column methods**, focusing on IORBO versus SBO. Here, it is important to carefully read the captions and interpretation scheme: when the statistical test shows “++” and the win rate is, for example, 0.623, this indicates that the *row method* (e.g., IOR-CD) is significantly better than the *column method* (e.g., SBO-V) with a win rate of 0.623. Following this interpretation across the tables, IORBO consistently outperforms SBO, both in terms of statistical significance and aggregated win rates.
>
> In summary, the proposed loss function is significantly better than the vanilla loss function when considered through the comprehensive and statistically rigorous lens (Tables 5–7), and IORBO consistently improves upon SBO when methods are compared row-to-column (Tables 8–9). We kindly ask the reviewer to revisit the tables and captions with this interpretation in mind.
>
> We hope this clarification resolves the concerns and we are happy to expand further if needed.
>
>
> **R1: Clarify better the connections between contributions: Improve the flow of the paper so that the three contributions (loss, benchmarking, IORBO) are better integrated and motivated rather than appearing as isolated components.**
>
> Please see W1.
>
>
>
> **R2: Correct theoretical statements: Rephrase or clarify inaccurate claims such as "neural networks cannot capture correlations."**
>
> Please see W4.
>
>
>
> **R3: Reframe the loss function: Clearly describe the proposed loss as an auxiliary term, not the main optimization objective.**
>
> Please see W5.
>
>
>
> **R4: Revise the related work section: Expand coverage to include prior work on: Handling heterogeneous tabular data and Bayesian optimization methods in generative modeling.**
>
> Please see W7.
>
>
>
> **R5: Clarify the novelty of the ML augmentation task: Either justify its novelty or acknowledge prior work that uses real + synthetic data to improve ML performance.**
>
> Please see W6.
>
>
> **R6: Fix citation for KL Divergence: Use the original reference [4] above.**
>
> We have corrected the citation for KL Divergence to reference the original source [4], as requested.
>
>
> **R7: Address clarity issues in methodology: Avoid redundancy in Sections 3.2, 3.4, and 6. Clarify when IORBO is introduced and used.**
>
> Please see W2.
>
>
> **R8: Evaluate and report IORBO cost: The paper claims the cost is negligible, but this should be justified with empirical or computational analysis.**
>
> Thank you for raising this important point regarding the computational cost of IORBO. We recognize that our initial statement of "negligible cost" required empirical validation.
>
> To address this, we conducted timing experiments comparing IORBO against standard SBO across different numbers of evaluated metrics (1, 10, 30, and 100) over 50 iterations each.
>
> We have added Figure 6 and the following text to the Results and Discussion section (highlighted in blue text):
>
> > **Bayesian optimization computational cost.** Figure 6 compares the per-iteration computational cost of IORBO and SBO across different numbers of evaluated metrics. Each iteration measures the time required to update the surrogate model (see Algorithms 1 and 2). The analysis shows that both IORBO and SBO have the same asymptotic computational complexity, scaling linearly with the number of metrics, i.e., $O(M)$. As shown in the timing comparison, execution times for both methods largely overlap across metric counts of 1, 10, 30, and 100, with observed differences remaining negligible. Thus, IORBO achieves its improved optimization performance without incurring additional computational cost in practice.

---

### Review · Reviewer_kBgG · 2025-07-26

**Summary Of Contributions:**

In this paper, the authors propose a new method to leverage generative models in the context of tabular data. They introduce additional loss regularizations — correlation- and distribution-based — that improve the quality of the generated synthetic data samples compared to other approaches used for tabular data. Namely, in addition to the generative loss function, they match correlations and distribution moments between real and generated data.

They also propose a convenient method to perform Bayesian optimization in the context of multi-dimensional targets, namely when one tries to optimize several metrics at once. The core idea is that instead of considering absolute metric values for a set of parameters, we can compare them in terms of relative rank value among each other, which makes the optimization targets adapt during each optimization step.

Finally, they propose a comprehensive evaluation setup for synthetic data-based tabular models, such as statistical similarity of generated data to real data, and ML performance of the model trained on synthetic data.

**Audience:**

Yes

**Claims And Evidence:**

Yes

**Requested Changes:**

* Clarify the unifying motivation behind the three contributions. Explain how loss regularization, IORBO-based optimization, and the evaluation framework work together.
* Discuss the limitations and scope of the correlation- and distribution-based losses beyond tabular data. Address why they are particularly suited for tabular data, and under what conditions they could generalize (e.g., to embeddings of other modalities).
* Provide experimental insights on scalability by including at least one experiment on a larger-scale tabular dataset (or synthetic proxy). Alternatively, discuss how the approach is expected to behave as the data size grows.

**Strengths And Weaknesses:**

**Strengths**

* As generative models for tabular data gain traction, addressing their robustness is both timely and highly relevant.
* The proposed method is conceptually simple, with broad applicability across various tabular data models.
* The paper is clear and easy to follow.

**Weaknesses**

* The paper lacks a clear unifying theme. Its three contributions — loss regularization, hyperparameter tuning, and evaluation — feel somewhat disconnected, with limited explanation of how they relate to each other. For example, the Bayesian optimization component could apply to any model involving multiple metrics, without any specific link to tabular data.
* In theory, the proposed correlation- and distribution-based losses could be applied to data types beyond tabular data. The main reason this might not work well for more structured data — such as images — is that the underlying distributions in pixel space are too complex, making correlations less meaningful. I suggest the authors clarify this point in the paper to better justify why their method is specifically suited to tabular data. Additionally, these losses could potentially be applied to image or text embeddings generated by (pre)trained encoders, which might extend the applicability of the method to other data modalities. A broader discussion would be appreciated.
* The experiments are conducted on relatively small tabular datasets, whereas real-world applications often involve millions of samples. This is a crucial limitation, as the size of the training data can significantly impact the behavior and performance of generative models. It also raises the question of how the proposed approaches scale and perform as the dataset size increases — an aspect not addressed in the current evaluation.

---

> ### Author Response · Authors · 2025-09-30
> **Reply to Reviewer kBgG [1]**
>
> **W1: The paper lacks a clear unifying theme. Its three contributions — loss regularization, hyperparameter tuning, and evaluation — feel somewhat disconnected, with limited explanation of how they relate to each other. For example, the Bayesian optimization component could apply to any model involving multiple metrics, without any specific link to tabular data.**
>
> Thank you for raising this important point. We agree that, at first glance, the three contributions could appear distinct. However, the unifying theme of our work is improving the robustness and utility of tabular generative models across their full lifecycle (training → tuning → evaluation). The motivation for each component arises directly from limitations in existing approaches:
>
> -   Loss regularization: Current tabular deep generative models (DGMs) often fail to preserve correlations and higher-order statistics, which reduces the fidelity of generated data. Our correlation- and distribution-aware loss addresses this gap by explicitly capturing richer distributional properties, including both continuous and discrete variables.
> -   Hyperparameter tuning: Standard sequential Bayesian optimization (SBO) is not well-suited for the tabular generation setting, where models must be tuned against multiple heterogeneous metrics. This often leads to biased or unstable outcomes. Our IORBO method addresses this by providing a robust, fair approach to multi-metric hyperparameter optimization.
> -   Existing evaluation practices are fragmented and narrow, often focusing on a small subset of metrics. This makes it difficult to compare models fairly or assess robustness. Our unified evaluation framework broadens this scope and ensures rigorous, consistent assessment across diverse datasets.
>
> By addressing these three gaps, our contributions form a cohesive pipeline for advancing tabular generative modeling, rather than isolated techniques. While each of the proposed components could in principle be applied more broadly, their integration is especially valuable in the tabular domain, where all three challenges co-occur.
>
> To make this unifying theme more explicit, we have revised the title, abstract, introduction, and conclusion. In specific, the revised title now reads:
> > A Unified Framework for Tabular Generative Modeling: Loss Functions, Benchmarks, and Improved Multi-objective Bayesian Optimization Approaches
>
> We have also expanded the introduction(highlighted in magenta in the revised draft):
> > [...]
> >
> > Yet tabular data poses unique challenges that resist direct transfer of techniques from other domains. Unlike images or text, tabular data lacks clear structure and contains mixed continuous and discrete variables with complex interactions, imbalances, and non-linear relationships. Recent hybrid approaches have explored combining diffusion processes with flow-based models and gradient-boosted trees to boost synthesis fidelity on tabular benchmarks (Jolicoeur-Martineau et al., 2024; Zein & Urvoy, 2022). However, these methods still rely on unguided likelihood or adversarial objectives and do not explicitly enforce key statistics such as feature correlations or higher-order moments. This mismatch between tabular data complexity and current generative approaches cascades across the entire modeling lifecycle. Existing deep neural network (DNN)-based generative models often struggle to reliably capture correlations and other statistical dependencies in tabular data—sometimes failing to approximate even basic statistics such as the mean and variance—particularly in limited-data settings (Xu et al., 2019). Current approaches to improve downstream machine learning (ML) analyses focus primarily on addressing data imbalance (Xu et al., 2019; Sun et al., 2023; Zhao et al., 2021) while neglecting the equally crucial role of feature distributions and correlations.
> >
> > This mismatch between tabular data complexity and current generative approaches cascades across the entire modeling lifecycle. Existing deep neural network (DNN)-based generative models often struggle to reliably capture correlations and other statistical dependencies in tabular data—sometimes failing to approximate even basic statistics such as the mean and variance—particularly in limited-data settings (Xu et al., 2019). Current approaches to improve downstream machine learning (ML) analyses focus primarily on addressing data imbalance (Xu et al., 2019; Sun et al., 2023; Zhao et al., 2021) while neglecting the equally crucial role of feature distributions and correlations.

---

> > ### Author Response · Authors · 2025-09-30
> > **Reply to Reviewer kBgG [2]**
> >
> > > The cascade extends to hyper-parameter optimization. While Bayesian optimization (BO) is widely used, standard approaches like standard Bayesian optimization (SBO) are ill-suited to aggregating the multiple heterogeneous metrics required for synthetic data evaluation. Combining metrics with different ranges and units, such as classification accuracy and regression error, via simple averaging can overweight individual objectives and yield suboptimal parameter selections.
> > Finally, the cascade undermines evaluation itself, where rigorous assessment remains fragmented. Existing methods often suffer from limited evaluation scopes that focus on narrow metric subsets, making it difficult to assess model performance across the complexities of diverse datasets. This evaluation gap obscures whether apparent improvements reflect genuine advances or artifacts of selective testing.
> > >
> > > The central problem is the absence of a unified framework that addresses tabular generative modeling across its full lifecycle: training, hyper-parameter tuning, and evaluation. Current approaches tackle these stages independently, missing opportunities for integrated solutions that could amplify improvements at each step.
> > To address this gap, we propose a comprehensive unified framework that tackles training, hyper-parameter tuning, and evaluation as interconnected challenges. First, we introduce a novel correlation- and distribution-aware loss function for DGMs designed to enforce statistical properties that existing generative models fail to capture reliably. Second, we develop iterative objective refinement Bayesian optimization (IORBO), which aggregates multiple evaluation metrics through ranking to resolve inconsistencies caused by metrics with different units or scales. Third, we establish a comprehensive benchmarking framework that evaluates synthetic data across twenty datasets using statistical, regression, and classification metrics. By integrating these components, we create a unified pipeline where training improvements and robust hyper-parameter tuning work in concert with rigorous evaluation. The tight coupling between training, tuning, and evaluation improves statistical fidelity, robust optimization, and benchmarking rigor across diverse datasets.
> > >
> > > [...]
> >
> > And in the Conclusion:
> >
> > > We presented a unified framework for tabular generative modeling that integrates training, tuning, and evaluation. While tightly integrated, each component is also designed to be applied independently, making the framework modular. [...]

---

> ### Author Response · Authors · 2025-09-30
> **Reply to Reviewer kBgG [3]**
>
> **W2: In theory, the proposed correlation- and distribution-based losses could be applied to data types beyond tabular data. The main reason this might not work well for more structured data — such as images — is that the underlying distributions in pixel space are too complex, making correlations less meaningful. I suggest the authors clarify this point in the paper to better justify why their method is specifically suited to tabular data. Additionally, these losses could potentially be applied to image or text embeddings generated by (pre)trained encoders, which might extend the applicability of the method to other data modalities. A broader discussion would be appreciated.**
>
> We agree that the correlation- and distribution-aware loss is general and could be applied well beyond tabular data. However, tabular datasets provide a context in which this loss is particularly effective, because feature-level correlations and marginal distributions constitute the primary informative signals. In contrast, raw correlations in structured data such as pixel or text space are often less meaningful due to strong local or hierarchical structure. We have clarified in the manuscript that while extending these losses to embeddings from other modalities (e.g., images or text) is promising, this is beyond the scope of the current work.
>
> We have added a dedicated subsection at the end of Section "A Correlation- and Distribution-Aware Loss Function" to clarify the scope (highlighted in magenta in the revised draft):
>
> > *Scope and applicability.* The proposed correlation- and distribution-aware loss functions are specifically designed for tabular data, where feature-level correlations and marginal distributions carry the most informative signals. While the losses themselves are general, applying them directly to raw pixels or text is less meaningful due to strong local structure and sequential dependencies. Nonetheless, the framework could be adapted to work on learned embeddings from other modalities, such as image or text encoders, which provide a structured representation suitable for correlation- and distribution-based regularization.
>
> Additionally, we added a note in the Discussion section on possible extensions:
>
> > While our work focuses on tabular datasets, we note that the proposed losses could potentially be applied to embeddings derived from images or text. Such embeddings capture feature-level structure in a way that preserves meaningful correlations and distributions, making them suitable targets for correlation- and distribution-aware regularization. Exploring this direction represents a promising avenue for future research.

---

> ### Author Response · Authors · 2025-09-30
> **Reply to Reviewer kBgG [4]**
>
> **W3: The experiments are conducted on relatively small tabular datasets, whereas real-world applications often involve millions of samples. This is a crucial limitation, as the size of the training data can significantly impact the behavior and performance of generative models. It also raises the question of how the proposed approaches scale and perform as the dataset size increases — an aspect not addressed in the current evaluation.**
>
> We appreciate the reviewer's concern regarding the scalability of our proposed methods to larger datasets. While our experiments include datasets ranging from small to moderately large scales—such as the Credit dataset with approximately 277,000 rows, the Diabetes with 234,245 rows—we acknowledge that real-world tabular data can encompass millions of samples.
>
> The primary limitation in extending our evaluations to such scales stems from computational constraints rather than inherent algorithmic issues. Specifically, our correlation- and distribution-aware loss functions exhibit linear scaling with respect to the number of features and batch size, ensuring computational efficiency. Similarly, the IORBO introduces only negligible overhead, as its ranking-based refinement involves lightweight operations that do not significantly increase runtime. We anticipate that our approach will perform robustly on larger datasets, with the losses continuing to enforce statistical fidelity without prohibitive costs, though empirical validation on massive scales would require substantial resources beyond what is currently available to us.
>
> To address this, we have revised the manuscript to emphasize the range of dataset sizes tested and to include a dedicated discussion on expected scaling behavior. We have added the following text at the end of the subsection describing the datasets:
>
> > Our benchmarking framework evaluates the proposed methods across twenty diverse real-world tabular datasets, spanning a range of scales to assess performance under varying conditions. These include small-scale datasets with thousands of rows, as well as mid- to large-scale ones, such as the Credit dataset with approximately 277,000 rows. This selection allows us to test the methods on datasets representative of practical scenarios, though we note that even larger datasets (e.g., millions of rows) are common in industrial applications and warrant future exploration.
>
> And added the following new paragraph immediately before the conclusion or future directions:
>
> > A key limitation of our current evaluation is that it focuses on datasets up to approximately 277,000 rows, whereas real-world tabular applications often involve millions of samples. This limitation is primarily due to computational time constraints rather than the scalability of the methods themselves. Specifically, training DGMs and performing extensive hyper-parameter optimization across multiple baselines becomes prohibitively time-consuming on very large datasets. In our super-computing environment, we impose a 7-day maximum training time per experiment, which effectively caps the dataset sizes we can explore. Theoretically, our correlation- and distribution-aware loss functions scale linearly with the number of features ($m$) and batch size ($B$), with time complexities of $\mathcal{O} (m^2 B)$ for the correlation term (due to pairwise computations) and $\mathcal{O} (mHB)$ for the distribution term (where $H$ is the number of moments). Since these losses operate on mini-batches rather than the full dataset, they remain efficient as dataset size grows. Similarly, IORBO adds negligible overhead, with its ranking and surrogate refitting steps scaling, which is minor compared to DGM training costs.  We expect our approach to behave robustly on larger scales: the losses should continue to enforce statistical fidelity effectively and may offer even greater benefits in high-volume scenarios where capturing complex dependencies is more challenging. Nonetheless, practical bottlenecks—particularly computational time, memory requirements for large batches, and the need for distributed training—may limit experiments on extremely large datasets.
>
> In addition, we have incorporated a new analysis of computational scaling. Figure 6 compares the per-iteration runtime of IORBO and SBO across different numbers of evaluation metrics, showing that both share the same asymptotic complexity ($O(M)$) and that their empirical runtimes largely overlap. This analysis reinforces that IORBO achieves its improved optimization performance without incurring additional computational cost, further supporting its applicability to larger-scale settings.

---

> ### Author Response · Authors · 2025-09-30
> **Reply to Reviewer kBgG [5]**
>
> **R1: Clarify the unifying motivation behind the three contributions. Explain how loss regularization, IORBO-based optimization, and the evaluation framework work together.**
>
> Please see W1.
>
>
> **R2: Discuss the limitations and scope of the correlation- and distribution-based losses beyond tabular data. Address why they are particularly suited for tabular data, and under what conditions they could generalize (e.g., to embeddings of other modalities).**
>
> Please see W2.
>
>
> **R3: Provide experimental insights on scalability by including at least one experiment on a larger-scale tabular dataset (or synthetic proxy). Alternatively, discuss how the approach is expected to behave as the data size grows.**
>
> Please see W3.

---

### Review · Reviewer_zMUy · 2025-09-16

**Summary Of Contributions:**

The paper targets the practical bottleneck in tabular synthetic data generation with deep generative models (DGMs) -- inability to capture the complexities of real-world tabular data. In order to mitigate this, authors propose novel additions to the loss function of DGMs: a correlation and distribution-aware regularizers for DGMs. Authors propose multi-objective aggregation strategy: Iterative Objective Refinement Bayesian Optimization (IORBO) and provide algorithm as well as the demonstration of its application to the toy example. Authors provide a benchmarking of different methods, illustrating the utility of the proposed approach, demonstrating the significant improvement of the fidelity of the synthetically generated data.

**Audience:**

Yes

**Claims And Evidence:**

Yes

**Requested Changes:**

The overall state of the paper is good, and I believe it meets all the requirements for acceptance. It would be beneficial to address the points raised in the Weaknesses/Questions section.

**Strengths And Weaknesses:**

### Strengths
1. Paper is written clearly and thoroughly.
2. Proposed approach is easy to apply for different DGMs
3. Deep evaluation: authors evaluated their approach on various datasets, demonstrating that IORBO provides advantages over the SBO
4. Theory for both correlation-aware and distribution-aware losses, providing bounds that could be used for improving training behaviour
5. Work demonstrates practical relevance for the tabular workflows and synthetic data generation for tabular data

### Weaknesses/Questions

1. Correlation loss ignores mixed-type dependencies. In the evaluation authors use Pearson correlation for continuous-continuous variables, and Cramer V for categorical-categorical variables, but what about continuous-categorical variables? It would be beneficial to clarify that in the manuscript.
2. Correlation-aware and distribution-aware loss prior works: I believe it would be very beneficial to the paper to discuss prior works in the area of statistical/correlation regularisers (for example [1]). Ideally, even using some of the similar approaches for the comparison, as currently comparisons are drawn with the standard baselines.

[1] Sun, Baochen, Jiashi Feng, and Kate Saenko. "Correlation alignment for unsupervised domain adaptation." Domain adaptation in computer vision applications. Cham: Springer International Publishing, 2017. 153-171.

---

> ### Author Response · Authors · 2025-09-30
> **Reply to Reviewer zMUy [1]**
>
> **W1: Correlation loss ignores mixed-type dependencies. In the evaluation authors use Pearson correlation for continuous-continuous variables, and Cramer V for categorical-categorical variables, but what about continuous-categorical variables? It would be beneficial to clarify that in the manuscript.**
>
> We thank the reviewer for highlighting the evaluation of continuous-categorical dependencies. Currently, our correlation-aware loss focuses on continuous-continuous (Pearson) and categorical-categorical (Cramer’s V) relationships. Continuous-categorical dependencies can be partially addressed by one-hot encoding the categorical variables before computing correlations. Additionally, our distribution-aware loss indirectly captures some mixed-type dependencies by aligning moments of both continuous and discrete features. We have clarified this in the manuscript and note that more explicit modeling of continuous-categorical correlations is an interesting direction for future work.
>
> In Section 3.3 (Evaluation), we have added a clarifying sentence (highlighted in orange in the revised draft):
>
> > While Pearson and Cramer’s V handle continuous-continuous and categorical-categorical correlations, continuous-categorical dependencies are not explicitly measured. These can be partially captured by one-hot encoding categorical variables or indirectly via the distribution-aware loss. Addressing this explicitly remains a direction for future work.
>
>
> **W2: Correlation-aware and distribution-aware loss prior works: I believe it would be very beneficial to the paper to discuss prior works in the area of statistical/correlation regularisers (for example [1]). Ideally, even using some of the similar approaches for the comparison, as currently comparisons are drawn with the standard baselines.**
>
> We thank the reviewer for highlighting the relevance of correlation- and distribution-aware regularizers. We have now discussed prior works, including Sun et al. (2017), which align second-order statistics in computer vision tasks. Our approach differs in several important ways: (1) it targets heterogeneous tabular data rather than visual domain adaptation; (2) it incorporates higher-order moments and handles both continuous and discrete features; and (3) it is applied across multiple generative models with evaluations on downstream statistical and ML tasks. These distinctions motivate our focus on empirical comparisons with the vanilla loss functions of each DGM, which isolates the impact of our proposed regularization. The related work section has been updated to clarify these differences (highlighted in orange in the revised draft):
>
> > Prior works in correlation and statistical regularization–most prominently CORAL by Sun et al. (2017)–focus on aligning second-order statistics (covariance) between source and target domains for unsupervised domain adaptation in computer vision. While effective for visual tasks, these methods typically assume continuous features and perform full covariance alignment, which can be costly both computationally and statistically in high-dimensional settings. By contrast, our correlation- and distribution-aware loss is tailored to tabular data with mixed continuous and discrete variables: it leverages variance- (diagonal covariance) based statistics for scalability, explicitly allows matching arbitrary higher-order moments (not just the second moment as in CORAL), and thus avoids the expense of estimating full covariance matrices while retaining the ability to capture richer distributional structure. Furthermore, our loss is integrated across multiple DGMs and evaluated on extensive downstream ML tasks and statistical metrics, providing theoretical guarantees alongside empirical improvements over standard baseline losses.

---

### Author Response · Authors · 2025-12-11
**Thank you**

Dear Action Editor and Reviewers,

Thank you for your time, careful reading, and constructive comments throughout the review process. Your feedback strengthened our paper and improved its clarity.

We appreciate the effort you invested in the evaluation.

Best regards,
Authors

---

### Decision · Action_Editor_qAxG · 2025-12-04

**Recommendation:** Accept as is

**Additional Comments:**

The reviewers are satisfied with the authors rebuttals, and the consensus is that the paper meets the TMLR acceptance criteria of being supported by accurate and convincing evidence, and of being of interest to the community. The reviewers had mixed opinions on the novelty or impact; however the work clearly meets the bar for TMLR.

While the reviewers are generally satisfied with the rebuttal, from the internal comments, there is still a sense that some of the proposed components feel somewhat detached from each other. The updated paper is clearly improved in this aspect, and so I am recommending accepting as-is; however this is something the authors might want to bear in mind in finalizing the camera-ready version.

**Audience:**

Yes

**Audience Explanation:**

Tabular generative modeling is of interest to many researchers and practitioners, and this addresses practically-important problems

**Claims And Evidence:**

Yes

**Claims Explanation:**

The reviewers agree that the evidence is accurate and convincing; there were only minor complaints about accuracy that were cleared up in the rebuttals.